# DELTA: Dual Consistency Delving with Topological Uncertainty for Active Graph Domain Adaptation

**Pengyun Wang**[1]      *pywang315@gmail.com*
**Yadi Cao**[2]      *yadicao95@gmail.com*
**Chris Russell**[1]      *chris.russell@oii.ox.ac.uk*
**Yanxin Shen**[3]      *shenyanxin5@163.com*
**Junyu Luo**[4]      *luo.junyu@outlook.com*
**Ming Zhang**[4]      *mzhang_cs@pku.edu.cn*
**Siyu Heng**[5]*      *siyuheng@nyu.edu*
**Xiao Luo**[6]*      *xiaoluo@cs.ucla.edu*

[1] *Oxford Internet Institute, University of Oxford*
[2] *Department of Computer Science and Engineering, University of California, San Diego*
[3] *School of Economics, Nankai University*
[4] *School of Computer Science, Peking University*
[5] *Department of Biostatistics, School of Global Public Health, New York University*
[6] *Department of Computer Science, University of California, Los Angeles*

**Reviewed on OpenReview:** *https://openreview.net/forum?id=P5y82LKGbY*

## Abstract

Graph domain adaptation has recently enabled knowledge transfer across different graphs. However, without the semantic information on target graphs, the performance on target graphs is still far from satisfactory. To address the issue, we study the problem of active graph domain adaptation, which selects a small quantitative of informative nodes on the target graph for extra annotation. This problem is highly challenging due to the complicated topological relationships and the distribution discrepancy across graphs. In this paper, we propose a novel approach named Dual Consistency Delving with Topological Uncertainty (DELTA) for active graph domain adaptation. Our DELTA consists of an edge-oriented graph subnetwork and a path-oriented graph subnetwork, which can explore topological semantics from complementary perspectives. In particular, our edge-oriented graph subnetwork utilizes the message passing mechanism to learn neighborhood information, while our path-oriented graph subnetwork explores high-order relationships from substructures. To jointly learn from two subnetworks, we roughly select informative candidate nodes with the consideration of consistency across two subnetworks. Then, we aggregate local semantics from its K-hop subgraph based on node degrees for topological uncertainty estimation. To overcome potential distribution shifts, we compare target nodes and their corresponding source nodes for discrepancy scores as an additional component for fine selection. Extensive experiments on benchmark datasets demonstrate that DELTA outperforms various state-of-the-art approaches. The code implementation of DELTA is available at https://github.com/goose315/DELTA.

## 1 Introduction

Graph data is widely applied in various real-world scenarios, for example in social networks (Guo & Wang, 2020; Zhang et al., 2022b), academic networks (Tang et al., 2008; West et al., 2016), transportation net-

---

*Corresponding authors.

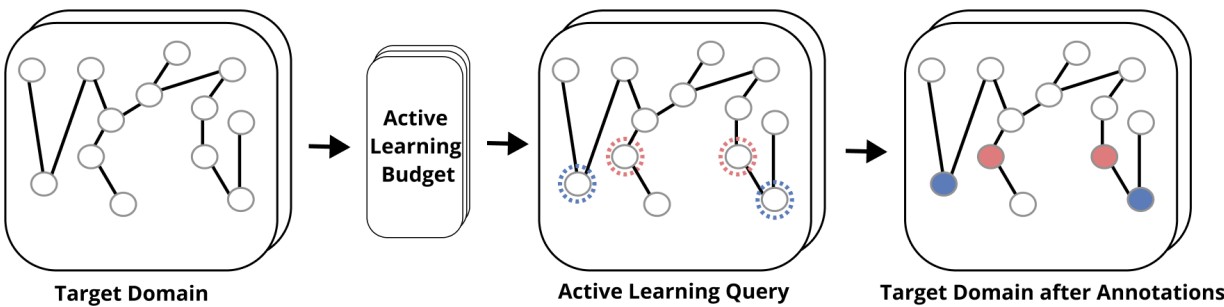

Figure 1: The problem setting of active learning for graph domain adaption. We have a fixed budget and aim to find the most informative nodes for data annotation.

works (Li et al., 2022; Jin et al., 2020), biological networks (Liu et al., 2024; Shen et al., 2021), compound networks (Sun et al., 2022; Cai et al., 2022), and drug discovery (Li et al., 2024; Wu et al., 2022b). In the collection of graphs, variations in the standards and timing often cause two graphs in the same domain to exhibit different node attributes and edge structures (Zhu et al., 2021; Pilancı & Vural, 2020).

Towards this end, unsupervised graph domain adaptation has received ever-increasing attention in the field of data mining and graph machine learning. Given a partially labeled source graph and a fully unlabeled target graph, graph domain adaptation narrows the distributional differences between the source and target graphs, enabling the model trained on the source domain to better adapt to graph data in the target domain, thereby improving task performance in the target domain (Hedegaard et al., 2021; Li et al., 2022; Cai et al., 2024). Current approaches to graph domain adaptation primarily usually involve adversarial learning (Qiao et al., 2023) for domain alignment, minimizing the distance between node representations (Wu et al., 2024), or filtering out irrelevant information between the source and target graphs through the information bottleneck strategies (Qiao et al., 2024).

However, the performance on the target graph is still far from satisfactory for unsupervised domain adaptation methods due to the absence of labels in the target graph (Cai et al., 2024). Due to constraints such as labeling costs, it is not feasible to extensively label the nodes on the target graph. Although pseudo-labeling strategies have been developed to address label scarcity issues in target graphs, they still fail to provide accurate supervision signals for the target graph, resulting in potential error accumulation (Qiao et al., 2024; Guo et al., 2022; Qiao et al., 2023). To reduce the annotation cost, we study the problem of active graph domain adaptation (see Figure 1), which aims to acquire the most valuable true labels for the target graph under a limited labeling budget, thereby maximizing the performance improvement on the target graph in a cost-effective manner.

Despite significant advancements in graph domain adaptation research, several challenges still need to be considered when applying active learning to graph domain adaptation (Li et al., 2022; Wang et al., 2023a; Shen et al., 2023a). *Firstly*, though extensive active learning methods (Hsu & Lin, 2015) have been proposed for images and texts, they focus on independent and identically distributed (i.i.d.) data. In contrast, graph data exhibit complex and high-order topological relationships, such as the number and density of nodes and edges, the heterogeneity of node degree distribution, and topological structure. These complicated factors greatly enhance the difficulties of identifying informative nodes on the target graph. *Secondly*, the source and target graphs have huge distribution discrepancies, which could deteriorate domain adaptation and result in biases in uncertainty estimation as well (Li et al., 2022; Qiao et al., 2024; 2023).

In this paper, we address the aforementioned challenges by introducing Dual Consistency Delving with Topological Uncertainty (DELTA) for active graph domain adaptation. The core of our DELTA is to explore graph topological data from complementary views for information-rich candidate nodes in the target graph. In particular, our DELTA consists of an edge-oriented graph subnetwork and a path-oriented graph subnetwork. Our edge-oriented subnetwork utilizes the message passing mechanism to learn topological semantics implicitly while our path-oriented subnetwork aggregates information from different paths explicitly.

Then, we combine two subnetworks by measuring the inconsistency across these subnetworks of nodes on the target graph for coarse selection. Furthermore, we measure the uncertainty by combining node degrees with K-hop subgraphs to learn from local topological semantics for each node. To mitigate the distribution shifts, we also calculate the discrepancy scores by comparing target nodes and their corresponding source nodes. In the end, we combine both uncertainty scores and discrepancy scores for fine selections.

We validate the effectiveness of DELTA through extensive experiments on benchmark datasets, comparing it with state-of-the-art approaches. We also demonstrate our advantages qualitatively through t-SNE visualization. The main contributions of this study are as follows:

- We study the problem of active graph domain adaptation and benchmark the performance of recent state-of-the-art approaches in transfer learning scenarios.

- We propose DELTA, a novel and cost-effective active learning sampling strategy that explores topological semantics from complementary edge and path perspectives, incorporating subgraph-level uncertainty and degree-weighted attribute discrepancy between the source and target graphs.

- We conduct extensive experiments on popular graph transfer learning benchmarks, demonstrating that the proposed DELTA performs better than state-of-the-art active learning approaches.

## 2 Related Work

**Active Learning on Graphs.** Active learning has been extensively studied in the field of computer vision (Hsu & Lin, 2015), but limited studies have applied active learning to graph-oriented deep learning (Cai et al., 2017; Gao et al., 2018; Chen et al., 2019; Hu et al., 2020; Cui et al., 2022; Zhang et al., 2022c; Yu et al., 2024; Zhang et al., 2022a). Earlier research generally does not consider the topological structure features of graphs. Still, it effectively combines node uncertainty and representativeness (Cai et al., 2017; Hu et al., 2020), where uncertainty is measured by information entropy, and representativeness is measured by information density and graph centrality (Gao et al., 2018; Chen et al., 2019). Many of these methods employ multi-armed bandit mechanisms to identify the weights of active learning strategies (Cai et al., 2017; Gao et al., 2018; Chen et al., 2019; Hu et al., 2020). More recent studies combine active learning strategies with reinforcement learning, formalizing active learning as a Markov decision process (Hu et al., 2020; Cui et al., 2022; Zhang et al., 2022c; Yu et al., 2024). They use measures such as PageRank centrality and predictive entropy to evaluate the informativeness and uncertainty of nodes (Zhang et al., 2022c) and combine metrics like KL divergence to assess the information value of nodes (Cui et al., 2022). However, although there are studies applying active learning to GNNs, most are based on a single graph. Few studies focus on applying this technique to graph transfer learning, which is more challenging compared to single-graph classification tasks (Shi et al., 2024). This study aims to fill this gap by conducting active learning across graphs and innovatively selecting nodes on the target graph that exhibit significant differences from the source graph. By leveraging active learning for labeling, we reduce information interference and distributional discrepancies in cross-graph learning, aligning with the specific characteristics of graph domain adaptation tasks.

**Graph Domain Adaptation.** Existing methods for graph domain adaptation can be categorized into three main streams (Shi et al., 2024): First, methods that enhance node embeddings on the source graph to improve performance on the target graph. This is achieved through adversarial loss functions (Yang et al., 2022; Yuan et al., 2023; Wu et al., 2022a; Yin et al., 2023; Qiao et al., 2023; Zhang et al., 2021) or by modifying the message passing mechanisms of GNNs (Wu et al., 2020; Dai et al., 2022; Shen et al., 2023b). Second, methods that focus on better adapting the knowledge learned from the source graph domain to the target graph domain. This is achieved through data augmentation (Jiang et al., 2020; Qiao et al., 2024), spectral methods (Pilancı & Vural, 2020; You et al., 2023), spatial methods (Guo et al., 2022; Wu et al., 2023), or pseudo-label alignment (Yin et al., 2022; Song et al., 2020). Third, methods that directly utilize information from the target graph to build models (Wang et al., 2023a;b; Shen et al., 2023a), using the probability distribution of predicted labels on the target graph (Wu & Rostami, 2023; Wang et al., 2023a) and semantic information (Wang et al., 2023b; Shen et al., 2023b). However, there is still a lack of methods that explicitly utilize graph topological information, while existing approaches only explore semantic

information from a single view, whether edge, path, or node. Current methods also rarely explicitly utilize the attribute distribution differences between the nodes in the source and target graphs, with most existing methods addressing this through distribution alignment loss functions (Sun et al., 2015; Cai et al., 2024). Furthermore, few methods directly address label sparsity in both the source and target graphs. Most existing studies assume a high proportion of labeled nodes in the source graph (Pilancı & Vural, 2020; Zhang et al., 2021; Guo et al., 2022; Wu et al., 2023; Sun et al., 2015; Cai et al., 2024). As a related problem, graph out-of-distribution detection aims to identify nodes that do not belong to a training distribution Song & Wang (2022); Gong & Sun (2024); Bao et al. (2024). In comparison with these aforementioned approaches, our DELTA enriches the labels of the target graph through active learning and innovatively utilizes an edge-oriented graph subnetwork and a path-oriented graph subnetwork to explore topological semantics from complementary perspectives.

**Active Learning for Domain Adaptation.** Existing research has not fully explored the integration of active learning with domain adaptation (Zhan et al., 2021; 2022; Han et al., 2023). ALDA uses the best classifier learned from the source domain as an initial hypothesis in the target domain and adjusts weights to query labels (Rai et al., 2010). Furthermore, the JO-TAL utilizes Maximum Mean Discrepancy (MMD) to measure marginal probability distribution differences between source and target domain samples, selecting instances from the unlabeled target domain dataset to minimize this discrepancy (Chattopadhyay et al., 2013). Additionally, AADA combines Domain-Adversarial Neural Networks (DANN) with a sample selection strategy based on Importance Weighted Empirical Risk Minimization (IWERM) to achieve domain-invariant feature learning (Su et al., 2020). CLUE builds upon AADA by further integrating predictive entropy to measure information content and using weighted k-means clustering to group similar target instances (Prabhu et al., 2021). Recent studies include EADA and ADCD. EADA selects and annotates the most informative unlabeled target samples by utilizing free energy, achieving a combination of inter-domain feature and instance uncertainty (Xie et al., 2022). ADCD improves active learning in domain adaptation by combining protocol scoring, domain discriminator scoring, and cosine difference scoring (Menke et al., 2024). In summary, there are currently no specialized studies based on active learning for graph domain adaptation. Given its significant practical implications, research in this direction is necessary (Shi et al., 2024; Redko et al., 2020). The proposed DELTA framework aims to fill this gap by comprehensively considering the complexity of graph topological structures and the distributional discrepancies in cross-graph learning.

# 3 Preliminaries

## 3.1 Problem Definition

Consider two graphs, one as the source graph $\mathcal{G}^s = (V^s, E^s, \mathbf{X}^s, Y^s)$ and one as the target graph $\mathcal{G}^t = (V^t, E^t, \mathbf{X}^t, Y^t)$. Here, $V^s$ and $V^t$ represent the sets of nodes in the source and target graphs, respectively, $E^s$ and $E^t$ represent the sets of edges in the source and target graphs, respectively, $\mathbf{X}^s$ and $\mathbf{X}^t$ donate the feature matrix of nodes from the source and target graphs, and $Y^s$ and $Y^t$ represent the sets of node class labels in the source and target graphs, respectively. The source graph and target graph have a serious domain gap but share the same label space, with only a small portion of $Y^s$ being labeled and $Y^t$ being completely unlabeled (Wilson & Cook, 2020). The objective of active graph domain adaptation is to find the most valuable nodes for labeling in the target graph $\mathcal{G}^t$ within a fixed budget of $k$, assign them annotations $Y^t$, and perform graph domain adaptation using a graph neural network model shared between the source and target graphs (Hsu & Lin, 2015; Xie et al., 2022; Rai et al., 2010).

## 3.2 Graph Domain Adaptation

We briefly present the typical framework of graph domain adaptation, a semi-supervised learning framework that has been widely applied (Dai et al., 2022; Yin et al., 2023; Qiao et al., 2024). Graph domain adaptation aims to ensure that node embeddings output by the feature extractor have similar distributions across different domains and ultimately label the unlabeled target nodes. Therefore, a classifier on node embeddings would generate predictions from the shared conditional distributions across the two domains. In formulation,

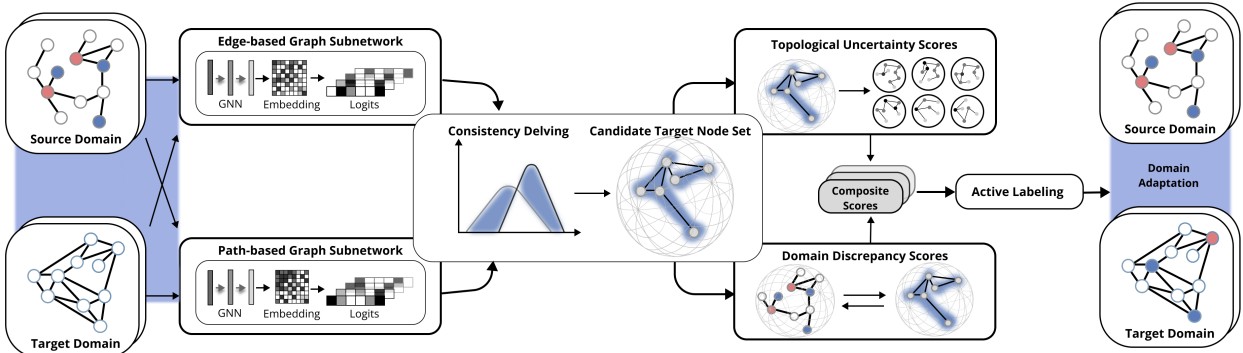

Figure 2: Overview of the proposed DELTA framework. DELTA first captures complementary topological semantics from the target graph via edge-oriented and path-oriented subnetworks. It then identifies candidate nodes with semantic inconsistencies between these subnetworks. DELTA computes topological uncertainty using degree weighting and K-hop subgraphs, and domain gap between the source and target graphs for informative node selection.

given the source and target embedding matrix $\mathbf{Z}^s$ and $\mathbf{Z}^t$, respectively, we have:

$$L = L_{Sup}(\mathbf{Z}^s, Y^s) + \lambda L_{DA}(\mathbf{Z}^s, \mathbf{Z}^t), \tag{1}$$

where $L_{Sup}$ denotes the cross-entropy objective on the labeled source graph $\mathcal{G}^s$, and $L_{DA}$ denotes the domain adaptation loss, such as the adversarial learning objective $L_{DA}$ (Qiao et al., 2024):

$$L_{DA} = \min_{\theta_d} \max_{\phi_d} \left[ \mathbb{E}_{i \in V^s} \log D(\mathbf{z}_i^s) + \mathbb{E}_{j \in V^t} \log(1 - D(\mathbf{z}_j^t)) \right]. \tag{2}$$

$D(\cdot)$ denotes a domain discriminator that predicts whether the input embedding is from the source graph or the target graph. $\mathbf{z}_i^s$ and $\mathbf{z}_j^t$ denotes the node embeddings of node $i \in V^s$ and $j \in V^t$, respectively. $\theta_d$ and $\phi_d$ denote the network parameters of the feature extractor and the classifier, respectively.

## 4 Methodology

### 4.1 Overview

In this work, we propose a new approach named DELTA for active graph domain adaption. DELTA consists of three main components as follows. (1) **Dual Graph Subnetwork with Consistency Delving**, which consists of an edge-oriented graph subnetwork and a path-oriented graph subnetwork trained on both the source and target graphs. Then, we explore complementary neighborhood information and high-order relationships from the target graph, resulting in informative candidate nodes denoted as $\mathcal{T}$. (2) **Topological Uncertainty Measurement**, which explores both target graph node degrees and K-hop subgraphs to aggregate local entropy, thereby capturing the topological uncertainty of the target graph nodes. (3) **Domain Discrepancy Measurement**, which calculates the distance between target nodes and their corresponding source labeled nodes, thus overcoming potential distribution shifts. By aggregating topological uncertainty and domain discrepancy scores on $\mathcal{T}$, we select the target graph nodes to be labeled in a single round for better convenience following (Sener & Savarese, 2017). The overview of the proposed method is illustrated in Figure 2.

### 4.2 Dual Graph Subnetwork for Consistency Delving

In our DELTA, we adopt a dual graph subnetwork framework that operates simultaneously on both the source and target graphs to explore graph semantics from complementary edge-centric and path-centric perspectives. Since the target graph is entirely unlabeled, while only a portion of the source graph is labeled,

this allows us to leverage supervision signals from the labeled nodes in the source graph, enabling the model to learn domain distribution discrepancies between the two graphs, thereby acquiring more accurate topological information for the nodes in the target graph (Zhao et al., 2024; Zhang et al., 2024; Qiao et al., 2023). By leveraging complementary topological information, DELTA is able to better identify candidate nodes with more complicated topological information on the target graph, which can then be used for active learning.

### 4.2.1 Edge-oriented Graph Subnetwork

message passing neural networks have been adopted in a large number of graph machine learning tasks (Dai et al., 2022) based on neighborhood aggregation. Therefore, we first introduce an edge-oriented graph subnetwork, which aggregates information from the neighboring nodes of each node. The subnetwork can gradually expand the receptive field of the nodes at each layer of the network and thus capture more topological information (Kipf & Welling, 2016) in an implicit manner. The update rules of the subnetwork at layer $l$ can be expressed as follows:

$$\mathbf{N}_i^{(l-1)} = \text{AGGREGATE}\left(\{\mathbf{h}_i^{(l-1)} : j \in \mathcal{N}_i\}\right), \tag{3}$$

$$\mathbf{h}_i^{(l)} = \text{UPDATE}\left(\mathbf{h}_i^{(l-1)}, \mathbf{N}_i^{(l-1)}\right), \tag{4}$$

where $\mathbf{h}_i^{(l-1)}$ and $\mathbf{N}_i^{(l)}$ represents the node embedding and neighborhood embeddings at the $(l-1)$-th layer, respectively. The node representations at each layer are updated through a two-step process. First, the AGGREGATE function collects information from neighboring nodes. This step is typically achieved through matrix operations, which generate new feature representations. Then, the UPDATE function applies a nonlinear transformation to the aggregated information, producing the node features $\mathbf{h}_i^{(l)}$ for the current layer. After stacking $L_{edge}$ layer, the final node embedding for each node $i$ is denoted as $\mathbf{z}_{edge,i}$.

### 4.2.2 Path-oriented Graph Subnetwork

However, graph data contains high-order structure semantics, and our edge-oriented graph subnetwork is difficult to capture them (Ma et al., 2020; Ye et al., 2024). To solve this, we introduce a path-oriented graph subnetwork, which can explicitly capture the topological semantics of a graph from the perspective of path connections (Ma et al., 2020). In particular, we aggregate information by computing the weighted sum of all possible paths between nodes. These paths consider not only the direct connections between nodes but also the indirect connections and the overall structure of the paths. The path-oriented subnetwork can capture deeper relationships between nodes and the global topological features of the graph. The updated rule of the path-oriented subnetwork at layer $l$ can be expressed as follows:

$$\mathbf{H}^{(l)} = \sigma\left(\mathbf{M}^{-1/2}\sum_{n=0}^{L} e^{-\frac{E_n}{T}}\mathbf{A}^n \mathbf{M}^{-1/2}\mathbf{H}^{(l-1)}\mathbf{W}^{(l-1)}\right), \tag{5}$$

where $\sum_{n=0}^{L} e^{-\frac{E_n}{T}}\mathbf{A}^n$ represents the path aggregation and weighted summation, where $n$ denotes the path length ranging from 0 to $L$ (with $L$ set to 3). $\mathbf{A}^n$ is the $n$-th power of the adjacency matrix $\mathbf{A}$, indicating the adjacency relationships of nodes at a distance of $n$ hops in the graph. $e^{-\frac{E_n}{T}}$ represents the learnable path weight, determined by the path energy $E_n$ and the temperature coefficient $T$. $\mathbf{H}^{(l-1)}$ is the input feature matrix at layer $l-1$, $\mathbf{W}^{(l-1)}$ is the weight matrix at layer $l-1$, and $\sigma(\cdot)$ is the ReLU activation function. After stacking $L_{edge}$ layers, the final node embedding for each node $i$ is denoted as $\mathbf{z}_{path,i}$. The inverse square root of the normalization matrix $\mathbf{M}^{-1/2} = \text{Diag}(M_1^{-1/2}, \cdots, M_{|V|}^{-1/2})$ can be defined as:

$$M_i = \sum_{j=1}^{|V|}\sum_{n=0}^{L} e^{-\frac{E_n}{T}} A^n[i,j], \tag{6}$$

where $[i,j]$ returns the corresponding element of the matrix. When $n > 1$, our path-oriented subnetwork can explore the high-order path-based semantics embedded, which edge-oriented graph subnetwork cannot explore. The following theorem shows that our path-oriented subnetwork takes all paths with the same length equally but treats paths with different lengths inequally.

**Theorem 4.1.** *According to Eqn. 5 in the main text, the message passing process of our path-oriented graph subnetwork is influenced equally by the paths of the same length. Moreover, if $E_0 \neq E_1 \neq \cdots \neq E_L$, the paths of different lengths contribute differently.*

The proof can be found in Appendix E. From Theorem 4.1, our path-oriented graph subnetwork is more dependent on the path view instead of node attributes. Therefore, it can provide a different view from the edge-oriented subnetwork for topological modeling.

### 4.2.3  Consistency Delving for Informative Candidate Nodes

The dual graph subnetwork framework allows for the complementary integration of information for the target graph nodes, which are oriented on edges and paths, respectively. Therefore, appropriate aggregation of this complementary information is crucial, as we need to select target graph nodes that are rich in complicated edge and path topological information. Typically, target nodes with high inconsistency across subnetworks should have high uncertainty and carry complicated semantics. In our DELTA, instead of simply using nodes with inconsistent predicted classes, we calculate the Euclidean distance between the logit scores $\mathbf{s}_{edge,j} = \beta_{edge}(\mathbf{z}_{edge,j})$ and $\mathbf{s}_{path,j} = \beta_{path}(\mathbf{z}_{path,j})$ with two classifiers on the target graph and identify the set of coarse candidate nodes whose Euclidean distance exceeds a customizable consistency threshold $\gamma$ as the coarse candidate set $\mathcal{T}$, as follows:

$$d(\mathbf{s}_{edge,j}, \mathbf{s}_{path,j}) = \|\mathbf{s}_{edge,j} - \mathbf{s}_{path,j}\|_2, \tag{7}$$

$$\mathcal{T} = \{j \mid d(\mathbf{s}_{edge,j}, \mathbf{s}_{path,j}) > \gamma\}. \tag{8}$$

Here, Euclidean distance between two branches can indicate the inconsistency of logic between two branches. Moreover, if two predictions from different subnetworks are inconsistent, the predictions are sensitive to the architectures, which indicates the uncertainty of the predictions with rich values to be labeled. Compared to using two identical subnetworks with different parameters to select nodes with inconsistent predictions, the dual graph subnetwork architecture allows us to obtain information about target graph nodes from two different views. This complementary approach enables the capture of more comprehensive node information and identifies target graph nodes with richer information and inconsistent predictions.

### 4.3  Topological Uncertainty Measurement for Node Selection

In active learning, uncertainty refers to the degree to which the model's predictions are less certain for different samples. By annotating nodes on the target graph where uncertainty is stronger, model performance can improve (Ma et al., 2024; Sharma & Bilgic, 2017; Fuchsgruber et al., 2024). Previous work mainly uses prediction entropy to measure uncertainty (Cai et al., 2017; Gao et al., 2018; Ren et al., 2022) for each node independently, while they neglect that the information of each node is highly related to its neighborhood. To tackle this, we propose topological uncertainty scores based on local subgraphs and degrees to capture topological semantics.

In particular, for each central node $j$ from $\mathcal{T}$ of the target graph, we first extract its K-hop subgraph and the logits of each node within it. Instead of directly averaging the logits of the K-hop subgraph, we compute the degree $d_m$ of each node in the K-hop subgraph and take the reciprocal as the weight $w_m = \frac{1}{d_m}$. The weighted K-hop logits for each node can be calculated as follows:

$$\hat{\mathbf{s}}_{edge,j} = \sum_{m \in \text{K-hop}(j)} w_m \mathbf{s}_{edge,m}, \tag{9}$$

$$\hat{\mathbf{s}}_{path,j} = \sum_{m \in \text{K-hop}(j)} w_m \mathbf{s}_{path,m}, \tag{10}$$

where $K$ is set to 2 empirically (Azabou et al., 2023). The reason behind using the reciprocal of a node's degree as the weight is that if a node has stronger connectivity (higher degree), its topological information can be inferred from the neighborhood with weaker importance (Fuchsgruber et al., 2024).

Then, we compute the topological uncertainty scores for each node in $\mathcal{T}$ as follows:

$$\phi_{\text{entropy}}(\hat{\mathbf{s}}_{edge,j}) = -\sum_{c=1}^{C} P(\hat{Y}_{jc} = 1 \mid \hat{\mathbf{s}}_{edge,j}) \log P(\hat{Y}_{jc} = 1 \mid \hat{\mathbf{s}}_{edge,j}), \tag{11}$$

$$\phi_{\text{entropy}}(\hat{\mathbf{s}}_{path,j}) = -\sum_{c=1}^{C} P(\hat{Y}_{jc} = 1 \mid \hat{\mathbf{s}}_{path,j}) \log P(\hat{Y}_{jc} = 1 \mid \hat{\mathbf{s}}_{path,j}), \tag{12}$$

where $\sum_{c=1}^{C} P(\hat{Y}_{ic} = 1 \mid \hat{\mathbf{s}}_{path,j})$ represents the probability that the weighted K-hop logits centered at node $j$ belong to category $c$.

Finally, each node $j$ from $\mathcal{T}$ of the target graph can obtain the topological uncertainty score $U_i$ from the dual graph subnetwork:

$$U_j = \phi_{\text{entropy}}(\hat{\mathbf{s}}_{edge,j}) + \phi_{\text{entropy}}(\hat{\mathbf{s}}_{path,j}). \tag{13}$$

By quantifying the topological uncertainty at the subgraph level, more comprehensive subgraph topological information can be obtained compared to node-based uncertainty. We also integrate the uncertainty scores from both the edge and path levels of the dual graph subnetwork, thereby aiding in the identification of more valuable nodes from different views.

### 4.4 Domain Discrepancy Measurement for Node Selection

Another obstacle in graph domain adaptation is the distribution differences between the source graph and the target graph (Yin et al., 2023; Yan et al., 2017). Even though several domain alignment approaches have been proposed to reduce the discrepancy, target nodes with high discrepancy with the source nodes would be more difficult for graph domain adaptation approaches (Qiao et al., 2024) to align and thus classify accurately. To this end, we propose domain discrepancy scores, which measure the attribute distance between the labeled nodes in the source graph and the candidate node set $\mathcal{T}$ in the target graph.

Specifically, given the attribute vector $\mathbf{x}_j^t$ of a target graph node $j \in \mathcal{T}$, and the attribute vectors $\mathbf{x}_i^s$ of nodes $i$ in the set of labeled source graph nodes $\mathcal{S}$, we compute the weighted average Euclidean distance between each $j$ and all labeled nodes in the set $\mathcal{S}$. The weights are determined by the degree $d_i$ of nodes in $\mathcal{S}$. The domain discrepancy score $D_i$ for each target graph node $j \in \mathcal{T}$ is defined as follows:

$$D_j = \frac{\sum_{i \in \mathcal{S}} d_i \cdot \|\mathbf{x}_j^t - \mathbf{x}_i^s\|_2}{|\mathcal{S}|}. \tag{14}$$

The reason behind using the degree of nodes in $\mathcal{S}$ as weights is that nodes with higher connectivity in the source graph are more influential in graph domain adaptation. In detail, they provide stronger supervision signals, thereby playing a more significant role in domain discrepancy scores. This design not only accounts for the differences in node attributes between the source and target graphs but also considers the topology of the source graph. Consequently, through the active learning process, our model focuses on target graph nodes less related to the source graph, ultimately enhancing the performance of graph domain adaptation.

### 4.5 Summarization

We combine the topological uncertainty scores $U_j$ and domain discrepancy scores $D_j$ of the target graph node $j \in \mathcal{T}$ to compute the composite score:

$$\mathcal{I}_j = U_j + D_j. \tag{15}$$

We first warm up both branches of subnetworks on the labeled source graph. Given the labeling budget $k$, we select the top $k$ nodes on $\mathcal{T}$ with the highest composite scores for active learning in a one-round manner, which selects the target graph nodes to be labeled in a single round as in (Sener & Savarese, 2017). The supervised loss objective is also applied when learning on the target graph as an additional signal, following

---

**Algorithm 1** Algorithm of DELTA

---

**Input:** Source graph $\mathcal{G}^s = (V^s, E^s, \mathbf{X}^s)$, labeled source graph set $\mathcal{S}$, target graph $\mathcal{G}^t = (V^t, E^t, \mathbf{X}^t)$, annotation budget $k$, consistency threshold $\gamma$, $K$ for K-hop subgraph.

**Output:** Selected target graph nodes for active learning.

 1: *// Dual Graph Subnetwork Training*
 2: Train edge-oriented subnetwork on $\mathcal{G}^s$ and $\mathcal{G}^t$, obtain logits $\mathbf{s}_{edge,j}$.
 3: Train path-oriented subnetwork on $\mathcal{G}^s$ and $\mathcal{G}^t$, obtain logits $\mathbf{s}_{path,j}$.
 4: *// Consistency Exploration*
 5: Calculate Euclidean distance between $\mathbf{s}_{edge,i}$ and $\mathbf{s}_{path,i}$ for each target node $j \in V^t$.
 6: Identify candidate node using Eqn. 8.
 7: *// Topological Uncertainty Measurement*
 8: **for** each node $j \in \mathcal{T}$ **do**
 9:     Extract the K-hop subgraph centered at $j$.
10:     Calculate topological uncertainty score $U_i$ using Eqn. 13.
11: **end for**
12: *// Domain Discrepancy Measurement*
13: **for** each node $j \in \mathcal{T}$ **do**
14:     Compute domain discrepancy score $D_j$ using Eqn. 14.
15: **end for**
16: *// Final Scores Computation*
17: **for** each node $v_i \in \mathcal{T}$ **do**
18:     Compute the composite score using Eqn. 15.
19: **end for**

---

existing graph domain adaptation frameworks (Qiao et al., 2024). The whole algorithm is summarized in Algorithm 1, and the computational complexity is provided in Appendix D.

In summary, the proposed DELTA framework provides a novel approach to active learning across graphs. It integrates complementary information from edge-oriented and path-oriented graph subnetworks and delves the inconsistency across two subnetworks for coarse candidate nodes. Then, it combines topological uncertainty scores with domain discrepancy scores for fine selection. This combination quantifies the uncertainty at the subgraph level within the target graph and the cross-domain distribution discrepancy between the target and source graphs, which significantly enhances the performance on the target graph with a minimal annotation budget.

# 5 Experiment

## 5.1 Experimental Settings

### 5.1.1 Datasets and Metrics

ArnetMiner is a system designed for exploring academic social networks (Tang et al., 2008). We select three citation networks from ArnetMiner: ACMv9 (A), Citationv1 (C), and DBLPv7 (D). Each node stands for a paper, and each edge denotes a citation relationship between the two papers. The node attribute is generated from the title followed by a bag-of-words model. The selected citation networks, ACMv9, Citationv1, and DBLPv7, each consist of five node categories. More details of the selected datasets can be found in Appendix A.

We adopt Micro-F1 and Macro-F1 as evaluation metrics (Hastie et al., 2004; Murphy, 2012). Unless otherwise specified, we report the average and variance of the results for the aforementioned metrics over five runs. Higher Micro-F1 and Macro-F1 values indicate better results.

Table 1: Summary of the performance of DELTA and baseline algorithms on each benchmark dataset, 5% nodes of source datasets are labeled, and 125 nodes of target datasets are labeled by active learning. The mean and variance over five runs are reported. A denotes acmv9, C donates citationv1, and D donates dblpv7. Macro donates Macro-F1, Micro donates Micro-F1. The best and second best are displayed in **bold** and underlined, respectively.

| Methods | A→C | | A→D | | D→C | | D→A | | C→A | | C→D | | Average | |
|---|---|---|---|---|---|---|---|---|---|---|---|---|---|---|
| | Macro | Micro | Macro | Micro | Macro | Micro | Macro | Micro | Macro | Micro | Macro | Micro | Macro | Micro |
| GIFI | 73.9±1.2 | 75.9±0.9 | 67.9±0.7 | 70.9±0.9 | 67.0±1.7 | 69.8±1.4 | 63.9±1.6 | 63.7±1.2 | 67.8±0.9 | 69.4±1.1 | 68.2±1.7 | 70.7±0.9 | 68.1 | 70.1 |
| SGDA | 74.5±0.9 | 76.5±0.8 | 68.9±1.0 | 70.9±1.1 | 70.1±0.6 | 71.2±0.3 | 65.2±2.3 | 66.5±1.3 | 68.9±0.5 | 70.1±0.7 | 68.4±1.4 | 70.8±1.5 | 69.3 | 71.1 |
| Random | 75.9±0.9 | 77.5±0.7 | 69.7±1.4 | 72.6±1.3 | 72.3±0.6 | 74.1±0.4 | 67.5±1.0 | 67.0±1.0 | 70.3±0.8 | 69.5±0.8 | 70.0±1.9 | 72.4±1.1 | 70.9 | 72.2 |
| GraphPart | 76.0±0.7 | 77.6±0.6 | **71.7±0.5** | **74.0±0.5** | 71.5±0.9 | 73.5±0.4 | 68.0±0.6 | 67.3±0.4 | 71.0±0.7 | 70.0±0.5 | 72.1±0.8 | 74.0±0.7 | 71.7 | 72.7 |
| AGE | 76.2±1.4 | 77.9±1.3 | 70.5±1.7 | 72.7±1.3 | 72.3±2.5 | 74.1±2.1 | 68.7±1.2 | 68.0±1.2 | 70.4±1.0 | 69.3±0.8 | 72.8±1.1 | 74.0±0.9 | 71.8 | 72.7 |
| ANRMAB | 74.9±1.0 | 76.8±0.8 | 69.6±1.0 | 71.9±0.7 | 70.7±1.3 | 73.0±0.8 | 66.6±0.7 | 66.0±0.6 | 70.0±0.6 | 69.0±0.6 | 69.5±1.5 | 71.8±1.1 | 70.2 | 71.4 |
| Dissimilarity | 76.3±1.4 | 77.9±1.3 | 70.0±2.3 | 72.6±1.9 | 73.3±1.9 | 75.1±1.6 | 68.1±0.7 | 67.5±0.6 | 69.7±0.8 | 68.8±0.5 | 71.0±2.3 | 72.8±1.5 | 71.4 | 72.4 |
| Degree | 74.5±0.9 | 76.3±0.8 | 68.7±1.2 | 71.3±1.1 | 71.4±1.0 | 73.5±0.7 | 66.3±1.0 | 65.9±0.8 | 68.4±0.4 | 67.7±0.5 | 70.7±0.6 | 72.2±0.3 | 70.0 | 71.1 |
| Density | 74.7±0.8 | 76.4±0.8 | 68.8±0.8 | 71.7±0.8 | 70.6±1.0 | 72.9±0.6 | 66.4±1.2 | 66.0±1.1 | 68.9±1.2 | 68.0±1.0 | 69.4±1.4 | 71.7±0.9 | 69.8 | 71.1 |
| Uncertainty | 76.0±1.2 | 77.7±1.2 | 71.0±1.5 | 73.2±1.2 | 71.8±1.7 | 73.9±1.5 | 68.5±1.7 | 67.9±1.4 | 71.2±0.1 | 70.3±0.2 | 70.8±1.3 | 72.5±0.6 | 71.6 | 72.6 |
| **Proposed** | **77.3±1.3** | **78.9±1.1** | 70.4±2.6 | 72.8±2.1 | **74.5±1.1** | **76.2±0.9** | **70.9±0.9** | **70.0±0.7** | **72.5±0.8** | **71.5±0.7** | **73.5±2.0** | **75.0±1.4** | **73.2** | **74.1** |

### 5.1.2 Baseline Algorithms

To validate the effectiveness of our DELTA, we compare it with a range of state-of-the-art baselines including GIFI (Qiao et al., 2024), SGDA (Qiao et al., 2023), GraphPart (Ma et al., 2023), AGE (Cai et al., 2017), ANRMAB (Gao et al., 2018), Dissimilarity (Ren et al., 2022), Degree (Cai et al., 2017), Density (Ren et al., 2022), Uncertainty (Settles & Craven, 2008) and Random. Random refers to labeling randomly selected samples with the budget. More details can be found in Appendix B.

### 5.1.3 Implementation Details

We use the GIFI model (Qiao et al., 2024) as the backbone model for graph domain adaption. The hidden channels are set to 512, out channels are set to 256, training epochs are 200, the dropout ratio is 0.1, and the Adam optimizer (Kingma & Ba, 2014) is utilized with a learning rate of 0.001, with a weight decay of 1e-4. We adopt GCN (Kipf & Welling, 2016) and PAN (Ma et al., 2020) to implement our two subnetworks, respectively. Following previous works (Sener & Savarese, 2017), we adopt a one-round setting in our experiments, which is more convenient in the real world with iteration number 1. For the GCN and PAN, both of them have two layers with 512-dimension node embeddings. The experimental configuration features a Linux server powered by NVIDIA A100 GPUs (80GB) and an Intel Xeon Gold 6354 CPU. The software environment includes PyTorch 1.11.0 (Paszke et al., 2019), PyTorch-geometric 2.5.3 (Fey & Lenssen, 2019), and Python 3.9.16.

## 5.2 Main Experimental Results

To quantitatively demonstrate the performance of our proposed DELTA, we report the results of DELTA and other baseline algorithms across six data combinations in Table 1, where 125 target graph nodes are labeled for active learning. From the results, we can observe that: (1) The proposed DELTA consistently outperforms the random selection method for active learning node selection, with an average lead of over 2% across the six data combinations. Note that random selection is quite strong with decent performance with a relatively large number of selected nodes. (2) Except for A→D, DELTA consistently outperforms other baseline algorithms, with the performance increasement range from 1.2% to 5% across the six data combinations. In D→C, D→A, and C→A, DELTA shows the largest margin, with an average lead of over 2%. (3) Baseline algorithms based on uncertainty, or combining uncertainty with other structural metrics, such as Uncertainty, GraphPart, AGE, and Dissimilarity, significantly outperform baselines using only structural metrics such as Degree and Density. (4) On average, Micro-F1 is greater than Macro-F1, indicating better classification performance for the larger classes, which reflects the issue of data imbalance. These results demonstrate the superior performance of the proposed DELTA, highlighting its advantage in reducing the annotation cost on the target graph. Notably, DELTA operates in a one-round manner for point selection on

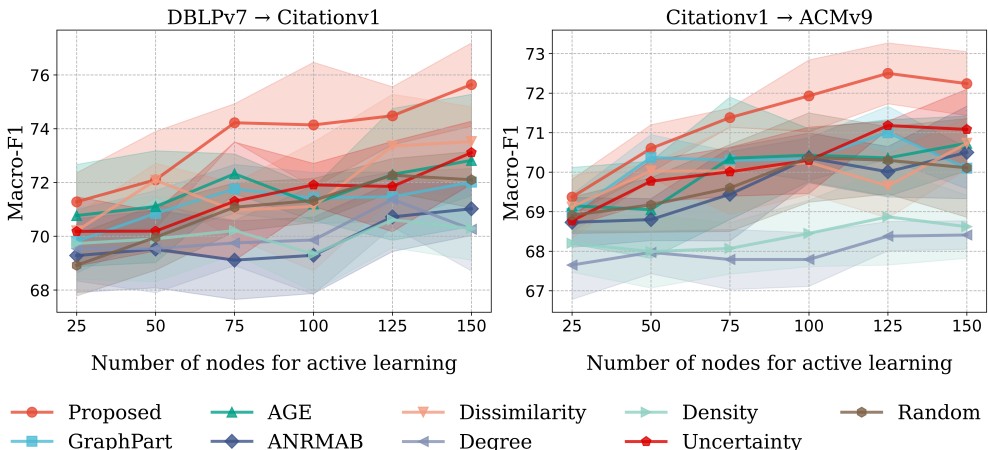

Figure 3: Performance of proposed method and baseline methods by active learning budget, averaged from 5 different runs. The Macro-F1 scores are plotted.

Table 2: The results of our ablation studies, in which 5% nodes of source datasets are labeled, and 50 nodes of target datasets are labeled by DELTA. The mean and variance over five runs are reported.

| Methods | A→C | | A→D | | D→C | | D→A | | C→A | | C→D | | Average | |
|---|---|---|---|---|---|---|---|---|---|---|---|---|---|---|
| | Macro | Micro | Macro | Micro | Macro | Micro | Macro | Micro | Macro | Micro | Macro | Micro | Macro | Micro |
| V1 | 75.0±1.1 | 76.9±1.0 | 69.1±1.4 | 71.8±0.9 | **72.3±1.1** | 74.0±1.2 | 67.5±0.9 | 67.0±0.8 | 70.5±0.4 | 69.7±0.5 | 69.5±1.4 | 71.8±0.8 | 70.7 | 71.8 |
| V2 | 68.3±8.8 | 72.2±8.9 | 66.8±0.9 | 72.5±1.3 | 70.2±1.1 | **74.8±1.6** | 63.6±5.9 | 65.1±3.2 | 68.3±1.6 | 67.6±1.5 | 63.0±5.2 | 68.6±3.0 | 66.7 | 70.1 |
| V3 | 75.2±0.4 | 77.1±0.3 | 69.2±0.6 | 71.8±0.4 | 71.7±1.1 | 73.5±0.6 | 66.4±0.9 | 66.0±0.9 | 69.5±0.4 | 68.8±0.5 | 69.7±1.4 | 71.7±1.1 | 70.3 | 71.5 |
| V4 | 75.5±1.2 | 77.3±1.1 | 69.3±0.8 | 72.3±0.3 | 70.7±1.5 | 73.2±1.4 | 66.6±1.3 | 66.3±1.2 | 70.1±1.3 | 69.3±1.1 | 70.2±1.6 | 72.3±1.0 | 70.4 | 71.8 |
| **Proposed** | **75.7±1.2** | **77.5±0.9** | **70.7±1.5** | **73.1±1.0** | 72.1±1.8 | 74.0±1.3 | **67.9±1.1** | **67.2±0.8** | **70.6±0.6** | **69.7±0.5** | **71.2±1.5** | **73.1±1.2** | **71.4** | **72.4** |

the target graph in active learning, while AGE, ANRMAB, and Dissimilarity involve iterative point selection, which indicates our DELTA achieves superior performance with lower computational costs.

To investigate the performance of DELTA and baseline algorithms under varying numbers of nodes selected for active learning, we visualized the results in Figure 3, where the number of selected nodes ranges from 25 to 150. From the results, it can be observed that: (1) Regardless of the number of nodes selected for active learning, DELTA consistently outperforms all baseline algorithms. (2) As the node number increases, DELTA's performance shows a continuous upward trend, and the performance gap between DELTA and other baseline algorithms tends to widen, which further proves the effectiveness of DELTA. For DELTA's performance across the six data combinations with active learning node counts of 100, 125, 150, 175, and 200, please refer to Table A2 in the Appendix, where the conclusions remain consistent.

## 5.3 Ablation Study

In this section, we introduce several variants to evaluate the effectiveness of different components in DELTA: (1) *V1* adopts two edge-oriented subnetworks with different parameters for inconsistency delving; (2) *V2* adopts two path-oriented subnetworks with different parameters for inconsistency delving; (3) *V3* removes domain discrepancy scores; (4) *V4* removes topological uncertainty scores. Table 2 presents the results of the ablation study, from which we can observe the following: (1) In most cases, the performance of DELTA consistently surpasses that of any of its ablated variants, with DELTA outperforming its variants by approximately 1% on average. (2) *V1* and *V2* indicate the use of two edge-oriented subnetworks or the use of two path-oriented subnetworks in our dual architecture. These variants do not realize complementary graph information acquisition, leading to less informative nodes in the candidate node set $\mathcal{T}$ compared to DELTA, thereby resulting in inferior performance in most cases. (3) *V3* refers to the use of only topological uncertainty scores without domain discrepancy scores. This variant underperforms DELTA across all datasets due to the failure to account for attribute differences between the source and target graphs in cross-graph learning, thus neglecting nodes with higher confusion during active learning. (4) *V4* represents the use of only domain

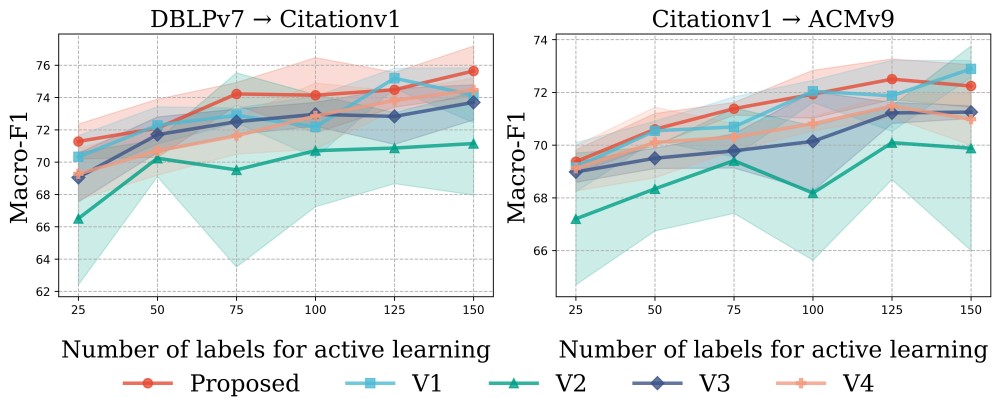

Figure 4: Performance comparison of ablation study.

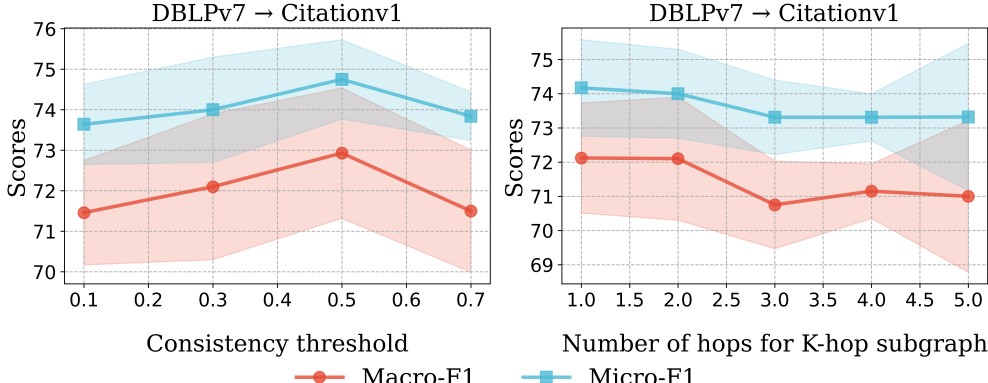

Figure 5: Performance of the proposed method by the consistency thresholds (left) and the number of hops for the K-hop subgraph (right), averaged from 5 different runs. In each sub-figure, the Macro-F1 and Micro-F1 scores are plotted. 50 nodes are selected for active learning.

discrepancy scores without topological uncertainty scores. This variant also shows inferior results compared to DELTA across all datasets because it fails to consider the uncertainty of the target graph nodes, which severely degrades the performance. Figure 4 further illustrates the ablation study results when varying the number of nodes used for active learning, and the conclusions are consistent with those in Table 2.

## 5.4 Parameter Sensitivity

In this part, we study the performance with respect to different consistency thresholds $\gamma$ and the values of $K$. $\gamma$ controls the consistency between the dual graph subnetworks, and $K$ is used to calculate topological uncertainty scores in $K$-hop subgraphs. We first conduct experiments by varying $\gamma$ within the parameter space $\{0.1, 0.3, 0.5, 0.7\}$ while keeping other parameters fixed. Then, we conduct experiments by varying $K$ within the parameter space $\{1, 2, 3, 4, 5\}$ while keeping other parameters fixed, as shown in Figure 6. We observe the following: (1) $\gamma$ exhibits an upward trend from 0.1 to 0.5, reaching its peak at 0.5, and then decreases to its lowest value at 0.7. The possible reason for this is that when $\gamma$ is too small, the consistency constraint on the dual graph subnetworks is too weak, leading to the inclusion of less informative nodes in $\mathcal{T}$. On the other hand, when $\gamma$ is too large, the consistency constraint becomes too strong, reducing the diversity of $\mathcal{T}$. (2) When $K$ is above 2, there is a slight downward trend in performance as $K$ increases, which might be due to the increase in noise within the topological uncertainty scores. This results in the subgraph size becoming too large, thereby diminishing the importance of the central node's logits, leading to decreased performance. Thus, we recommend setting $\gamma$ to 0.3 and $K$ to 2. Figures A1 and A2 in the Appendix provide further information with similar conclusions.

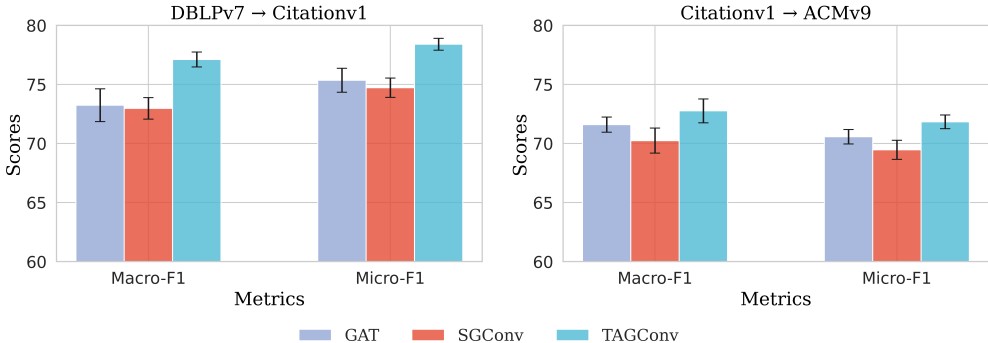

Figure 6: Performance of the proposed method by changing the edge-oriented backbones, averaged from 5 different runs. In each sub-figure, the Macro-F1 and Micro-F1 scores are plotted. 50 nodes are selected for active learning.

## 5.5 Further Analysis

In this subsection, we primarily investigate the generalizability of DELTA from two perspectives. First, we analyze the performance of DELTA when different backbones are employed in the edge-oriented subnetworks. Second, we compare the runtime of DELTA with baseline algorithms to assess its efficiency. Third, we utilize t-SNE to visualize the differences in the diversity of selected node distributions between the proposed DELTA and two classic baseline methods, Uncertainty (Settles & Craven, 2008) and Degree (Cai et al., 2017).

### 5.5.1 Performance on Different Edge-oriented Subnetworks

It is necessary to explore the generalizability of the edge-oriented graph subnetworks under different backbones. We replace the edge-oriented graph subnetworks with Graph Attention Networks (GAT) (Veličković et al., 2017), Simplifying Graph Convolutional Networks (SGConv) (Wu et al., 2019), and Topology adaptive graph convolutional networks respectively (TAGConv) (Du et al., 2017), as shown in Figure 6. These three types of encoders explore graph convolution operations from three representative perspectives, including adaptive weight allocation, efficiency optimization, and topological structures, respectively. We observe the following: (1) The best results are obtained when TAGConv is used as the first edge-oriented subnetwork, while the results are relatively poorer when GAT or SGConv is used as the edge-oriented subnetwork. (2) Overall, even with different backbones for the edge-oriented graph subnetwork, the proposed DELTA still demonstrates superior performance, which validates DELTA's strong generalizability and the effectiveness of achieving complementary information through different graph subnetworks. For more insights into the generalizability across additional datasets, please refer to Figure A3 in the Appendix, where the conclusions remain robust.

### 5.5.2 Performance vs Runtime

Figure 7 illustrates the trade-off between runtime and performance for the proposed DELTA compared to baseline algorithms. From the results, it can be observed that the runtime of DELTA is significantly shorter than that of AGE, ANRMAB, Dissimilarity scores, and GraphPart, but slightly longer than that of Degree, Density, and other remaining baselines. However, DELTA achieves the best performance. The potential reason is that AGE, ANRMAB, and Dissimilarity scores are iterative active learning algorithms with high time costs for each iteration (Cai et al., 2017; Gao et al., 2018; Ren et al., 2022), while GraphPart involves community clustering (Ma et al., 2023), which also requires a considerable amount of time. In contrast, the proposed DELTA takes only one-fifth of the time of AGE, ANRMAB, and Dissimilarity scores yet outperforms these methods. Figure A4 in the Appendix reports the runtime analysis across more datasets, where the conclusions remain robust.

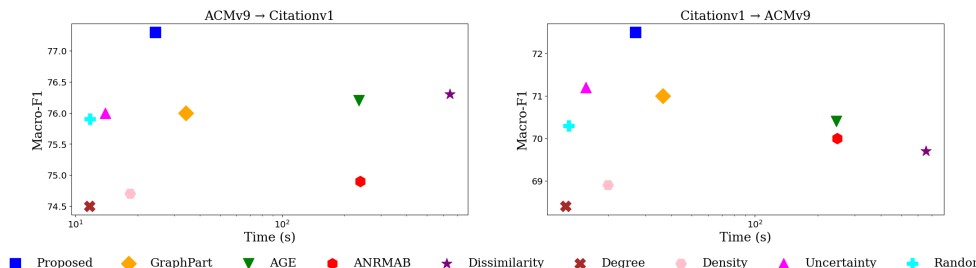

Figure 7: Performance vs. runtime of DELTA and baseline algorithms. 125 nodes are selected for active learning.

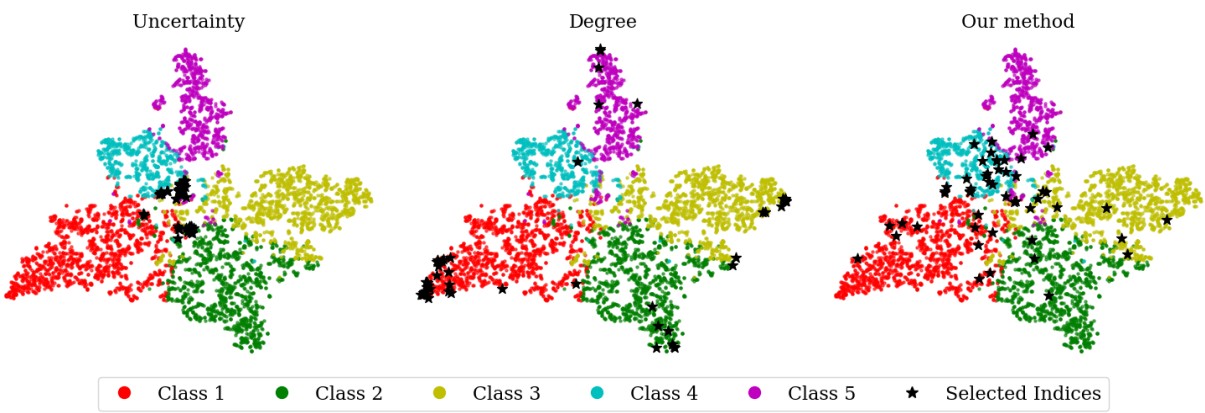

Figure 8: ACMv9→DBLPv7: visualization of logits scores output by the domain adaptation model on the target domain using t-SNE, with asterisks indicating the 50 nodes for active learning.

### 5.5.3 Visualization

Figure 8 shows the t-SNE visualization of the differences between the proposed DELTA and two classic baseline algorithms, Uncertainty (Settles & Craven, 2008) and Degree (Cai et al., 2017) in the ACMv9→DBLPv7 setting. From this, we can observe the following: (1) The nodes selected by the proposed DELTA cover each species category and are relatively evenly distributed across the target graph's node categories, highlighting the strong diversity of the nodes selected by DELTA. (2) In contrast, the nodes selected by Uncertainty and Degree are concentrated in 1 to 3 categories, with a highly uneven distribution. This contrast visually emphasizes the effectiveness and diversity of DELTA, as the richer the true labels of the nodes selected for active learning in the target graph, the easier it is to optimize the semi-supervised loss function in the target graph during graph domain adaptation. The more informative the labeled nodes in the target graph, the greater the improvement in the performance of graph domain adaptation. Figures A5 and A6 in the Appendix show t-SNE visualizations on additional datasets, where the conclusions remain consistent with those in Figure 8.

## 6 Discussion and Conclusion

Active learning for graph domain adaptation is both practical and important due to several key challenges and opportunities. First, annotating graph data is inherently expensive, since graph data is not independent and identically distributed (i.i.d.), and labeling nodes often requires domain expertise to account for graph structure, significantly increasing the cost (Yin et al., 2023; Hu et al., 2020). Second, the growing focus on data-efficient learning, driven by sustainability goals and the need to reduce carbon emissions, highlights the importance of maximizing the labeling budget's effectiveness in model training through active learning (Cai et al., 2017; Gao et al., 2018; Ren et al., 2022). Third, recent unsupervised domain adaptation methods demonstrate limited performance without labeled data in the target graph, where minimal but strategic

labeling can substantially improve results (Qiao et al., 2024; 2023; Prabhu et al., 2021). These factors underscore the practicality and necessity of active learning in this context.

Therefore, in this paper, we investigate the problem of active graph domain adaptation and propose a new approach named DELTA for this problem. Our DELTA consists of an edge-oriented graph subnetwork and a path-oriented graph subnetwork to explore topological semantics from complementary perspectives, and then selects target nodes with high inconsistency as candidate nodes. Then, DELTA combines both node degree and K-hop subgraphs to explore topological uncertainty for each node. It also calculates degree-weighted discrepancy scores to focus on target nodes differently from source nodes for fine selection. Extensive experiments on various benchmark datasets demonstrate the effectiveness of our DELTA. In future work, we will extend our DELTA to more generalized problems, such as open-set graph domain adaptation, and utilize large language models to mitigate the annotation burden.

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

## A Detailed Description of Datasets

Table A1 presents the descriptive statistics of acmv9, citationv1, and dblpv7. There are significant differences among the three graphs: acmv9 has the largest size (9,360 nodes and 15,602 edges), citationv1 has the highest average degree (1.691), and dblpv7 has the smallest graph size and the lowest average degree (5,484 nodes, 8,130 edges, average degree of 1.482). Regarding the node category ratios for the three graphs, the second category (Networking) is the most prevalent, while the fourth category (Information Security) is the least represented.

Table A1: The summary statistics of six graphs.

| Datasets | Node Number | Edge Number | Class Number | Average Degree |
|----------|-------------|-------------|--------------|----------------|
| ACMv9 | 9,360 | 15,602 | 5 | 1.667 |
| Citationv1 | 8,935 | 15,113 | 5 | 1.691 |
| DBLPv7 | 5,484 | 8,130 | 5 | 1.482 |

## B Baseline Algorithms

**GIFI.** GIFI is a state-of-the-art semi-supervised graph domain adaptation method, which employs variational information bottleneck to keep the crucial semantics in the graph data (Qiao et al., 2024).

**SGDA.** SGDA utilizes an adversarial manner with shift parameters to align the distribution across different domains (Qiao et al., 2023).

**GraphPart.** GraphPart first parities the target graph into communities. Then, it runs the K-means algorithm on each community and selects the node closest to the cluster center for labeling (Ma et al., 2023).

**AGE** AGE calculates the information entropy, information density, and graph centrality for each node in the target graph, linearly combines these metrics, and selects the node with the highest combined score for labeling (Cai et al., 2017).

**ANRMAB** ANRMAB uses a multi-armed bandit algorithm to weigh and combine the three metrics of AGE and selects the node with the highest combined score for labeling (Gao et al., 2018).

**Dissimilarity.** Dissimilarity scores build upon AGE by introducing the feature dissimilarity score (FDS) and structure dissimilarity score (SDS). These scores are linearly combined, and it labels nodes with the highest combined scores (Ren et al., 2022).

**Degree.** Selects the node with the highest degree centrality in the target graph for labeling (Cai et al., 2017; Ren et al., 2022).

**Density.** Runs K-means on the hidden representations of the target graph nodes and selects the node with the highest density score for labeling (Cai et al., 2017; Ren et al., 2022).

**Uncertainty.** Calculates the entropy of each node and selects the node with the highest entropy for labeling (Settles & Craven, 2008).

**Random.** Randomly selects nodes uniformly across the target graph for labeling.

# C   More experiments

In this section, we extend the experiments presented in the main text. First, we expand the comparison with baseline algorithms by using 100, 125, 150, 175, and 200 nodes for active learning to explore the effectiveness of the proposed DELTA. Then, we report the parameter sensitivity experiments, generalizability experiments, runtime vs performance analysis, and visualization analysis on the complete dataset combination.

## C.1   Performance Comparison

In Table A2, we report the comparison between the proposed DELTA and baseline algorithms when the number of nodes selected for active learning is set to 100, 125, 150, 175, and 200. From the results, we can observe the following: (1) On average, DELTA consistently outperforms all baseline algorithms. This further highlights the effectiveness of the proposed DELTA. (2) As the number of nodes selected for active learning increases, the performance gap of DELTA widens, from an average of 1-4% with 100 nodes to an average of 2-6% with 200 nodes. This indicates that DELTA performs better in medium to large-scale active learning tasks. (3) As the number of selected nodes increases, baseline methods based on community partitioning, such as GraphPart, and combined metrics, such as AGE, ANRMAB, and Dissimilarity, gradually fall behind simpler baseline methods based on single metrics, such as Uncertainty. The underlying reason is that as the number of selected nodes increases, the internal clusters within these methods become smaller, leading to a loss of node representativeness.

## C.2   Ablation Study

Table A3 presents additional ablation studies, where *V5* denotes the removal of the dual subnetwork and its consistency-based strategy for identifying informative candidate nodes, leaving only a single edge-oriented graph network. Similarly, *V6* represents the removal of the dual subnetwork and its consistency-based strategy, retaining only a single path-oriented graph network. From Table A3, we can observe that the proposed DELTA method still significantly outperforms *V6* and, on average, exceeds both *V5* and *V6*. On the A→C and D→C tasks, DELTA slightly lags behind *V5*. A possible explanation for this is that DELTA focuses on identifying nodes with strong prediction inconsistency between the two subnetworks, potentially overlooking some valuable nodes for active learning whose predictions are consistent across the subnetworks, thus causing DELTA to neglect them.

## C.3   Parameter Sensitivity

Figure A1 and Figure A2 report the results on the complete set of six dataset combinations for different values of $\gamma$ in the parameter space {0.1, 0.3, 0.5, 0.7} and $K$ in the parameter space {1, 2, 3, 4, 5}, respectively. From Figure A1, we can observe that the overall performance significantly declines when $\gamma$ exceeds 0.5, while the results are relatively better when $\gamma$ is between 0.3 and 0.5. This further validates our recommendation in the main text, suggesting that $\gamma$ be set between 0.3 and 0.5 to achieve a balance between the richness of topological complementary information and node diversity. From Figure A2, we can observe that DELTA achieves optimal performance when $K$ is set to 2, which is consistent with our suggestion in the main text. We recommend setting $K$ between 1 and 2 to balance the richness of subgraph information while preserving the contribution of the central node in the K-hop subgraph to topological uncertainty.

## C.4   Further Analysis

Figure A3 and Figure A4 respectively report the results of DELTA when varying GNN encoders on the complete set of six dataset combinations and the performance vs. runtime results on ACMv9→Citationv1, DBLPv7→ACMv9, and Citationv1→DBLPv7. The results in Figure A3 are consistent with the observations in the main text, where TAGConv performs the best when replacing the edge-oriented graph subnetwork. Overall, when changing edge-oriented encoders, the performance remains strong in comparison to most baseline algorithms. This further demonstrates the effectiveness of the proposed DELTA framework. From Figure A4, we can also observe the same conclusions as in the main text: the proposed DELTA's time cost

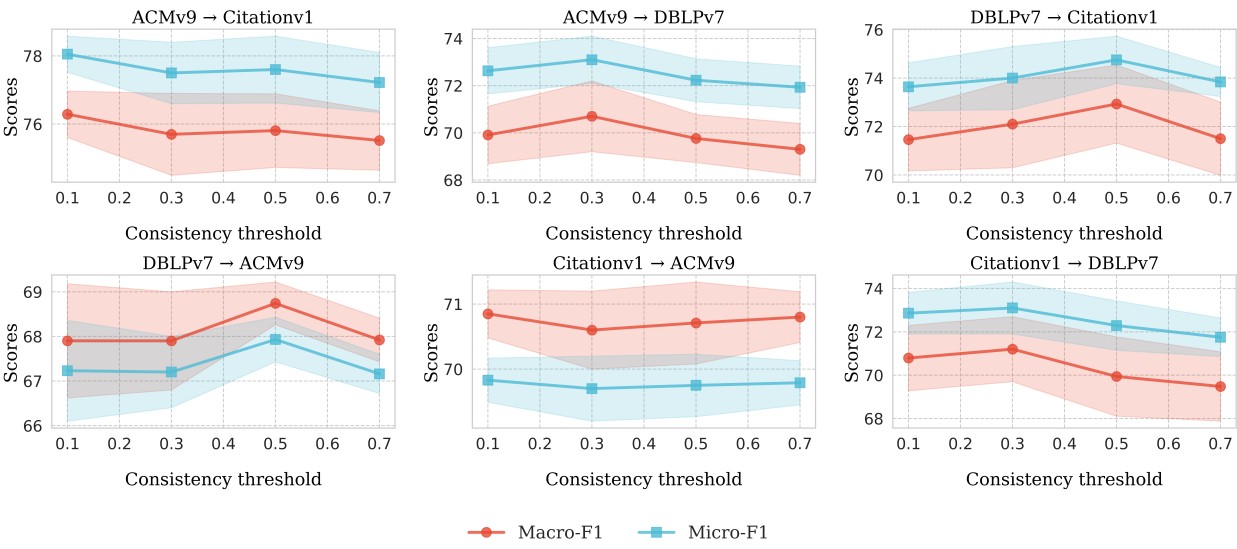

Figure A1: Performance of proposed method by the consistency thresholds, averaged from 5 different runs. In each sub-figure, the Macro-F1 and Micro-F1 scores are plotted, the number of hops is set to 2, and 50 nodes are selected for DELTA.

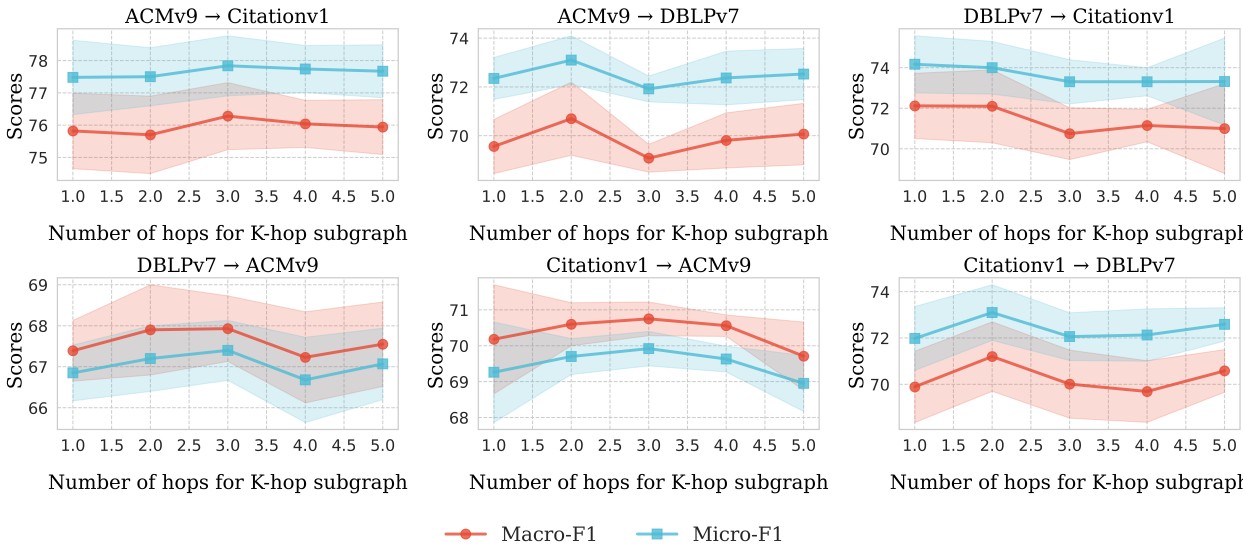

Figure A2: Performance of proposed method by the number of hops for K-hop subgraph, averaged from 5 different runs. In each sub-figure, the Macro-F1 and Micro-F1 scores are plotted, consistency thresholds are set to 0.3, and 50 nodes are selected for DELTA.

Table A2: Summary of the performance of active learning methods on each benchmark dataset, in which 5% nodes of source datasets are labeled, and 100, 125, 150, 175, and 200 nodes of target datasets are labeled by DELTA, respectively. The mean and variance over five runs are reported. A denotes acmv9, C donates citationv1, and D donates dblpv7. Macro donates Macro-F1, Micro donates Micro-F1.

| Methods | A→C | | A→D | | D→C | | D→A | | C→A | | C→D | | Average | |
|---|---|---|---|---|---|---|---|---|---|---|---|---|---|---|
| | Macro | Micro | Macro | Micro | Macro | Micro | Macro | Micro | Macro | Micro | Macro | Micro | Macro | Micro |
| *100 nodes are selected for active learning* | | | | | | | | | | | | | | |
| GraphPart | 76.0±0.3 | 77.8±0.2 | **72.1±0.8** | **74.4±0.8** | 71.4±0.9 | 73.3±0.9 | 67.8±0.3 | 67.2±0.1 | 70.3±0.5 | 69.4±0.5 | 71.4±0.8 | 73.7±0.5 | 71.5 | 72.6 |
| AGE | 76.4±1.3 | 78.0±1.2 | 69.7±1.9 | 72.1±1.4 | 71.2±0.6 | 73.3±0.6 | 68.2±1.0 | 67.7±0.9 | 70.4±0.7 | 69.5±0.5 | 71.3±1.1 | 73.1±0.6 | 71.2 | 72.3 |
| ANRMAB | 74.6±0.7 | 76.4±0.6 | 68.7±0.7 | 71.2±0.8 | 69.3±1.4 | 71.5±1.3 | 65.8±2.0 | 65.4±1.7 | 70.4±0.6 | 69.5±0.5 | 68.6±1.9 | 71.6±1.3 | 69.5 | 70.9 |
| Dissimilarity | 75.8±1.4 | 77.5±1.3 | 70.0±2.6 | 72.5±2.1 | 71.1±2.4 | 73.1±2.0 | 65.7±1.6 | 65.5±1.3 | 70.2±0.8 | 69.1±0.8 | **71.7±2.7** | 73.5±2.0 | 70.8 | 71.8 |
| Degree | 74.2±0.8 | 75.9±0.7 | 68.3±1.2 | 71.2±1.2 | 69.9±2.0 | 72.3±1.1 | 64.9±0.5 | 64.8±0.4 | 67.8±0.7 | 67.2±0.7 | 70.2±1.0 | 72.3±0.4 | 69.2 | 70.6 |
| Density | 74.2±0.7 | 76.2±0.8 | 69.8±0.9 | 72.4±1.0 | 69.3±1.4 | 71.8±0.9 | 65.6±0.7 | 65.2±0.8 | 68.5±0.8 | 67.8±0.7 | 69.3±1.3 | 71.8±0.7 | 69.5 | 70.9 |
| Uncertainty | 76.0±1.0 | 77.8±0.9 | 70.3±1.5 | 72.7±1.3 | 71.9±0.8 | 73.8±1.1 | 67.5±1.8 | 67.1±1.5 | 70.3±0.8 | 69.4±0.8 | 69.2±1.5 | 71.7±0.9 | 70.9 | 72.1 |
| Random | 75.3±0.7 | 77.0±0.7 | 69.6±1.6 | 72.5±1.4 | 71.3±0.8 | 73.2±0.7 | 67.1±0.6 | 66.6±0.5 | 70.4±1.1 | 69.4±1.0 | 68.9±1.2 | 71.6±0.5 | 70.4 | 71.7 |
| **Proposed** | **77.2±1.2** | **78.9±1.2** | 69.9±2.2 | 72.5±1.6 | **74.1±2.3** | **76.0±1.9** | **68.4±1.9** | **67.7±1.6** | **71.9±0.9** | **70.8±0.7** | 71.1±2.8 | 73.3±1.8 | **72.1** | **73.2** |
| *125 nodes are selected for active learning* | | | | | | | | | | | | | | |
| GraphPart | 76.0±0.7 | 77.6±0.6 | **71.7±0.5** | **74.0±0.5** | 71.5±0.9 | 73.5±0.4 | 68.0±0.6 | 67.3±0.4 | 71.0±0.7 | 70.0±0.5 | 72.1±0.8 | 74.0±0.7 | 71.7 | 72.7 |
| AGE | 76.2±1.4 | 77.9±1.3 | 70.5±1.7 | 72.7±1.3 | 72.3±2.5 | 74.1±2.1 | 68.7±1.2 | 68.0±1.2 | 70.4±1.0 | 69.3±0.8 | 72.8±1.1 | 74.0±0.9 | 71.8 | 72.7 |
| ANRMAB | 74.9±1.0 | 76.8±0.8 | 69.6±1.0 | 71.9±0.7 | 70.7±1.3 | 73.0±0.8 | 66.6±0.7 | 66.0±0.6 | 70.0±0.6 | 69.0±0.6 | 69.5±1.5 | 71.8±1.1 | 70.2 | 71.4 |
| Dissimilarity | 76.3±1.4 | 77.9±1.3 | 70.0±2.3 | 72.6±1.9 | 73.3±1.9 | 75.1±1.6 | 68.1±0.7 | 67.5±0.6 | 69.7±0.8 | 68.8±0.5 | 71.0±2.3 | 72.8±1.5 | 71.4 | 72.4 |
| Degree | 74.5±0.9 | 76.3±0.8 | 68.7±1.2 | 71.3±1.1 | 71.4±1.0 | 73.5±0.7 | 66.3±1.0 | 65.9±0.8 | 68.4±0.4 | 67.7±0.5 | 70.7±0.6 | 72.2±0.3 | 70.0 | 71.1 |
| Density | 74.7±0.8 | 76.4±0.8 | 68.8±0.8 | 71.7±0.8 | 70.6±1.0 | 72.9±0.6 | 66.4±1.2 | 66.0±1.1 | 68.9±1.2 | 68.0±1.0 | 69.4±1.4 | 71.7±0.9 | 69.8 | 71.1 |
| Uncertainty | 76.0±1.2 | 77.7±1.2 | 71.0±1.5 | 73.2±1.2 | 71.8±1.7 | 73.9±1.5 | 68.5±1.7 | 67.9±1.4 | 71.2±0.1 | 70.3±0.2 | 70.8±1.3 | 72.5±0.6 | 71.6 | 72.6 |
| Random | 75.9±0.9 | 77.5±0.7 | 69.7±1.4 | 72.6±1.3 | 72.3±0.6 | 74.1±0.4 | 67.5±1.0 | 67.0±1.0 | 70.3±0.8 | 69.5±0.8 | 70.0±1.9 | 72.4±1.1 | 70.9 | 72.2 |
| **Proposed** | **77.3±1.3** | **78.9±1.1** | 70.4±2.6 | 72.8±2.1 | **74.5±1.1** | **76.2±0.9** | **70.9±0.9** | **70.0±0.7** | **72.5±0.8** | **71.5±0.7** | **73.5±2.0** | **75.0±1.4** | **73.2** | **74.1** |
| *150 nodes are selected for active learning* | | | | | | | | | | | | | | |
| GraphPart | 76.5±0.6 | 77.9±0.5 | **71.9±1.3** | **74.3±1.2** | 72.0±0.5 | 73.7±0.6 | 68.3±0.5 | 67.8±0.4 | 70.1±0.5 | 69.1±0.5 | 71.7±0.9 | **74.3±0.3** | 71.7 | 72.8 |
| AGE | 75.4±1.1 | 77.1±1.0 | 70.3±2.3 | 72.7±1.7 | 72.8±2.5 | 74.7±1.8 | 68.8±0.9 | 68.1±0.9 | 70.7±0.7 | 69.7±0.7 | 71.0±2.8 | 73.1±1.5 | 71.5 | 72.6 |
| ANRMAB | 75.7±0.7 | 77.4±0.7 | 69.3±1.1 | 71.8±1.0 | 71.0±1.0 | 73.1±0.8 | 66.1±1.6 | 65.7±1.4 | 70.5±1.2 | 69.5±1.0 | 68.6±1.9 | 71.4±1.1 | 70.2 | 71.5 |
| Dissimilarity | 76.5±1.4 | 78.1±1.2 | 70.2±1.9 | 72.7±1.7 | 73.5±1.3 | 75.4±0.8 | 67.1±1.8 | 66.7±1.7 | 70.7±0.7 | 69.8±0.7 | 69.1±2.3 | 71.6±1.4 | 71.2 | 72.4 |
| Degree | 74.8±0.7 | 76.5±0.6 | 68.7±1.3 | 71.3±1.2 | 70.3±1.6 | 72.7±1.2 | 66.3±0.4 | 66.0±0.2 | 68.4±0.4 | 67.8±0.3 | 71.0±0.8 | 72.5±0.7 | 69.9 | 71.1 |
| Density | 75.2±0.7 | 76.9±0.6 | 70.0±1.1 | 72.8±0.8 | 70.3±1.2 | 72.6±0.5 | 66.0±1.4 | 65.6±1.2 | 68.6±0.8 | 67.8±0.8 | 70.3±1.6 | 72.6±1.0 | 70.1 | 71.4 |
| Uncertainty | 75.5±1.6 | 77.5±1.0 | 71.2±1.1 | 73.3±1.0 | 73.1±1.2 | 75.1±1.1 | 68.9±1.3 | 68.4±1.1 | 71.1±1.0 | 70.2±0.9 | 71.9±2.6 | 73.5±1.7 | 71.9 | 73.0 |
| Random | 76.2±0.8 | 77.8±0.6 | 70.5±1.5 | 73.4±1.5 | 72.1±1.1 | 74.0±0.7 | 67.8±0.8 | 67.2±0.7 | 70.1±1.2 | 69.2±1.2 | 70.4±1.9 | 72.8±1.2 | 71.2 | 72.4 |
| **Proposed** | **77.5±1.5** | **79.1±1.2** | 71.5±2.7 | 73.8±2.3 | **75.6±1.6** | **77.3±1.2** | **70.2±2.1** | **69.3±1.8** | **72.2±0.8** | **71.1±0.7** | 72.0±2.5 | 73.9±1.7 | **73.2** | **74.1** |
| *175 nodes are selected for active learning* | | | | | | | | | | | | | | |
| GraphPart | 76.7±0.8 | 78.2±0.7 | **73.0±0.5** | **75.2±0.4** | 71.7±0.8 | 73.5±0.6 | 68.0±0.7 | 67.2±0.7 | 70.9±0.7 | 70.1±0.7 | 71.9±1.1 | 73.9±0.7 | 72.0 | 73.0 |
| AGE | 76.2±1.6 | 77.9±1.4 | 70.5±2.6 | 72.9±2.0 | 73.1±2.5 | 74.8±2.2 | 66.4±2.1 | 66.1±1.8 | 70.5±1.0 | 69.5±0.9 | **73.6±0.8** | **74.7±0.7** | 71.7 | 72.6 |
| ANRMAB | 75.6±1.2 | 77.4±1.1 | 69.5±1.5 | 71.9±1.4 | 72.4±0.7 | 74.1±0.6 | 68.0±1.0 | 67.2±0.9 | 70.8±1.0 | 69.8±0.9 | 70.2±2.2 | 72.3±1.8 | 71.1 | 72.1 |
| Dissimilarity | 75.9±2.0 | 77.7±1.7 | 69.8±2.4 | 72.4±1.8 | **74.8±0.7** | **76.2±0.7** | 67.8±3.0 | 67.3±2.7 | 70.6±0.9 | 69.6±0.6 | 70.2±3.5 | 72.7±2.3 | 71.5 | 72.8 |
| Degree | 74.6±0.8 | 76.4±0.7 | 69.5±1.5 | 72.0±1.3 | 71.2±0.9 | 73.4±0.5 | 65.9±0.5 | 65.6±0.4 | 68.6±0.6 | 67.9±0.6 | 69.5±0.8 | 72.0±0.5 | 69.9 | 71.2 |
| Density | 75.0±0.9 | 76.7±0.8 | 69.7±2.0 | 72.4±1.7 | 69.6±1.3 | 72.2±0.8 | 65.8±1.0 | 65.5±0.9 | 68.8±0.5 | 68.1±0.5 | 70.1±2.0 | 72.4±1.4 | 69.8 | 71.2 |
| Uncertainty | 77.4±1.5 | 78.9±1.2 | 70.4±2.8 | 72.7±2.0 | 73.9±1.5 | 75.6±1.4 | 68.3±2.4 | 67.7±2.1 | 72.0±0.8 | 71.0±0.8 | 73.2±1.7 | 74.3±1.3 | 72.5 | 73.4 |
| Random | 76.3±1.2 | 78.0±1.1 | 69.8±2.3 | 72.8±1.9 | 72.2±1.6 | 74.1±1.4 | 67.0±1.0 | 66.6±0.9 | 71.1±0.6 | 70.1±0.4 | 70.7±1.2 | 73.2±0.7 | 71.2 | 72.4 |
| **Proposed** | **77.4±1.6** | **79.1±1.4** | 72.2±2.9 | 74.1±2.5 | 74.2±1.9 | 76.2±1.5 | **70.4±1.7** | **69.5±1.5** | **73.5±0.6** | **72.4±0.5** | 72.8±2.1 | 74.4±1.7 | **73.4** | **74.3** |
| *200 nodes are selected for active learning* | | | | | | | | | | | | | | |
| GraphPart | 76.7±0.5 | 78.3±0.4 | 72.5±1.4 | 74.7±1.0 | 71.8±1.4 | 73.8±1.1 | 68.2±0.9 | 67.4±0.8 | 71.3±0.8 | 70.2±0.7 | 72.4±1.1 | 74.2±0.9 | 72.2 | 73.1 |
| AGE | 75.9±1.7 | 77.6±1.6 | 71.0±2.0 | 73.2±1.6 | 72.8±1.7 | 74.4±1.6 | 66.8±0.8 | 66.6±0.9 | 70.2±1.2 | 69.2±1.1 | 71.5±2.6 | 73.2±1.7 | 71.6 | 72.5 |
| ANRMAB | 75.7±0.8 | 77.5±0.7 | 69.9±1.6 | 72.5±1.3 | 71.5±2.2 | 73.5±1.5 | 67.2±1.3 | 66.6±1.2 | 71.3±0.5 | 70.2±0.6 | 70.2±1.9 | 72.3±1.4 | 71.0 | 72.1 |
| Dissimilarity | 76.5±1.2 | 78.2±1.1 | 70.3±2.4 | 72.9±1.8 | 73.2±1.5 | 74.8±1.4 | 68.5±0.8 | 67.9±0.7 | 71.2±1.1 | 70.2±0.8 | 73.1±1.1 | 74.3±0.7 | 72.1 | 73.1 |
| Degree | 74.6±0.7 | 76.4±0.7 | 69.2±1.8 | 71.6±1.5 | 71.5±0.9 | 73.5±0.6 | 66.0±0.4 | 65.7±0.4 | 68.5±0.8 | 67.9±0.7 | 70.2±0.9 | 72.2±0.7 | 70.0 | 71.2 |
| Density | 74.7±0.7 | 76.6±0.6 | 69.6±1.6 | 72.4±1.3 | 69.1±2.1 | 71.8±0.9 | 65.5±1.2 | 65.3±1.0 | 69.1±0.4 | 68.3±0.5 | 71.1±0.8 | 73.1±0.6 | 69.9 | 71.3 |
| Uncertainty | 75.9±2.1 | 77.8±1.6 | 70.7±3.1 | 72.9±2.0 | 74.8±1.0 | 76.3±0.8 | 69.4±0.6 | 68.8±0.6 | 72.4±0.1 | 71.4±0.2 | 71.4±2.6 | 73.2±1.8 | 72.4 | 73.4 |
| Random | 76.5±0.9 | 78.1±0.9 | 71.6±1.2 | 74.0±0.8 | 72.5±1.0 | 74.3±1.1 | 68.2±0.9 | 67.5±0.8 | 70.8±1.4 | 69.9±1.0 | 70.0±2.0 | 72.6±1.5 | 71.6 | 72.7 |
| **Proposed** | **78.7±0.6** | **80.2±0.6** | **74.4±1.9** | **75.9±1.5** | 73.6±1.3 | 75.4±1.3 | **70.5±1.6** | **69.7±1.5** | **73.5±0.5** | **72.5±0.4** | **73.8±3.0** | **75.4±2.4** | **74.1** | **74.9** |

is only one-fifth to one-sixth that of AGE, ANRMAB, and Dissimilarity, while it only takes about 10 to 20 seconds more than other baseline algorithms. Nevertheless, DELTA achieves the best performance.

Figure A5 and Figure A6 respectively report the t-SNE visualization results of the proposed DELTA compared with Uncertainty and Degree centrality on ACMv9→Citationv1 and DBLPv7→ACMv9. From Figure A5 and Figure A6, we can observe the same conclusions as in the main text: DELTA achieves a more balanced and diverse selection of target graph nodes. In contrast, the nodes selected by Uncertainty and Degree centrality are concentrated in 1-2 categories, and this imbalanced node distribution significantly reduces

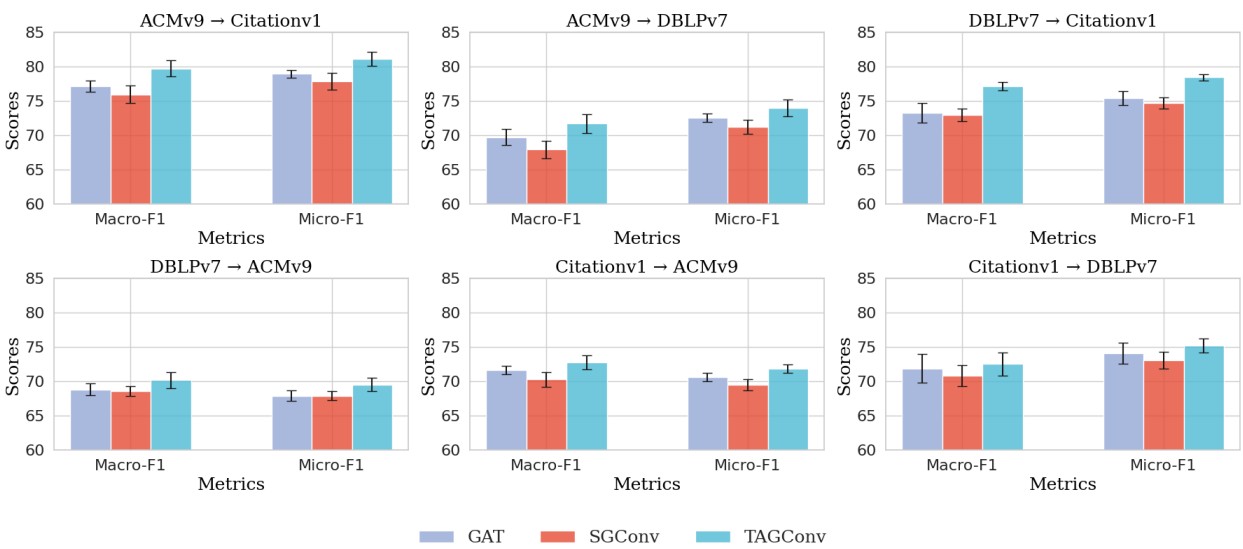

Figure A3: Performance of proposed method by changing the GNN backbones, averaged from 5 different runs. In each sub-figure, the Macro-F1 and Micro-F1 scores are plotted. 50 nodes are selected for DELTA.

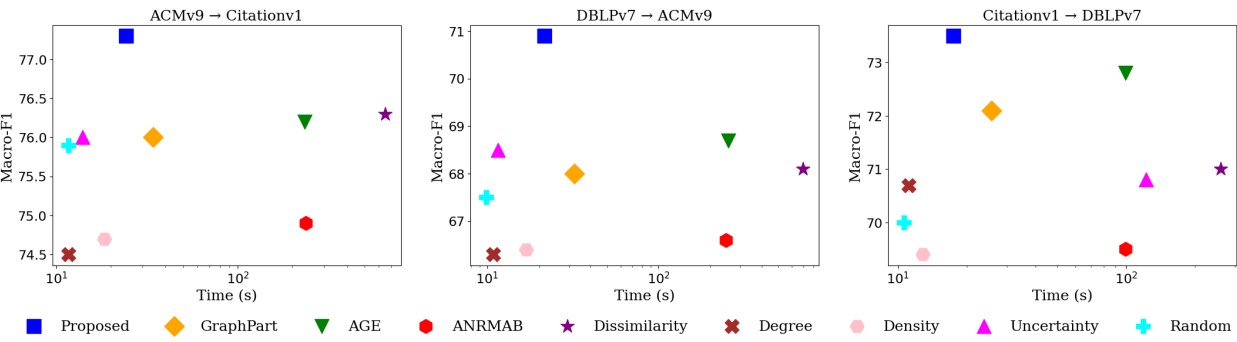

Figure A4: Performance vs. runtime of DELTA and baseline algorithms, in which 125 nodes are selected for DELTA.

Table A3: The results of our ablation studies, in which 5% nodes of source datasets are labeled, and 50 nodes of target datasets are labeled by DELTA. The mean and variance over five runs are reported.

| Methods | A→C | | A→D | | D→C | | D→A | | C→A | | C→D | | Average | |
|---|---|---|---|---|---|---|---|---|---|---|---|---|---|---|
| | Macro | Micro | Macro | Micro | Macro | Micro | Macro | Micro | Macro | Micro | Macro | Micro | Macro | Micro |
| V5 | 76.4±0.5 | 78.0±0.4 | 69.0±1.4 | 71.8±1.0 | 72.3±0.5 | 74.0±0.5 | 66.9±1.5 | 66.4±1.2 | 69.5±0.9 | 68.7±0.5 | 70.3±1.9 | 72.5±1.6 | 70.7 | 71.9 |
| V6 | 69.2±4.4 | 74.6±5.0 | 62.5±5.3 | 70.1±2.4 | 68.3±5.0 | 74.4±2.2 | 67.2±0.7 | 67.0±0.7 | 67.2±3.6 | 67.0±3.0 | 66.3±3.7 | 71.2±2.9 | 66.8 | 70.7 |
| **Proposed** | 75.7±1.2 | 77.5±0.9 | 70.7±1.5 | 73.1±1.0 | 72.1±1.8 | 74.0±1.3 | 67.9±1.1 | 67.2±0.8 | 70.6±0.6 | 69.7±0.5 | 71.2±1.5 | 73.1±1.2 | 71.4 | 72.4 |

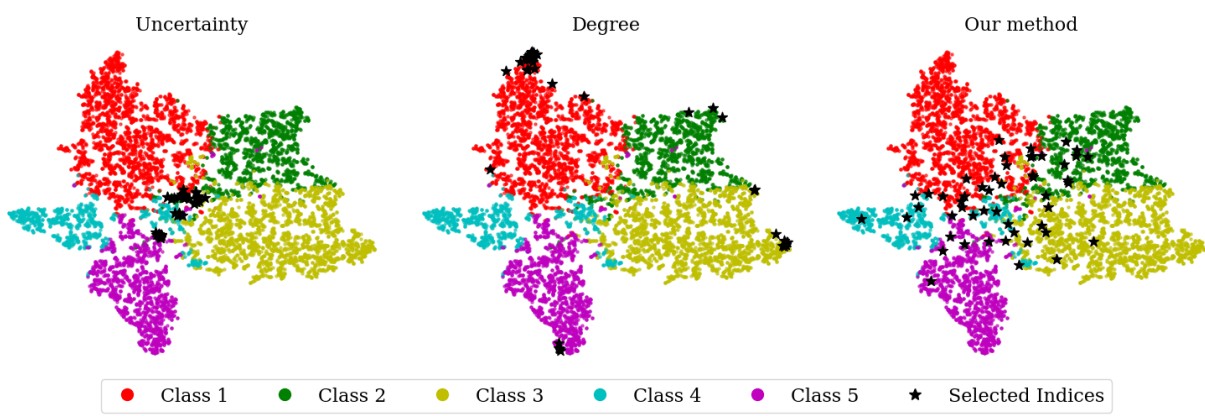

Figure A5: ACMv9→Citationv1: visualization of logits scores output by the domain adaptation model on the target domain using t-SNE, with asterisks indicating the 50 nodes for active learning.

the effectiveness of active learning. In comparison, DELTA's more uniform selection qualitatively provides evidence for the superior performance achieved by the proposed DELTA.

## D Computational Complexity Analysis

Assume $V$ represents the average node number in the input graph, $E$ represents the average edge number in the input graph, and $d$ is the hidden dimension. Calculating the K-hop subgraph has a time complexity of $O(E+V)$. Computing the K-hop subgraph for each node results in a total time complexity of $O(V(E+V))$. Averaging the logits for each subgraph node has a time complexity of $O(Vd)$. Calculating the softmax for logits and subgraph logits has a time complexity of $O(Vd)$.

Considering the above steps, the overall time complexity for the uncertainty computation process can be expressed as:

$$O(V(E+V) + Vd + Vd) = O(V(E+V) + 2Vd) = O(V(E+V+d)).$$

## E Proof of Theorem 4.1

*Proof.* We have

$$\mathbf{A}^n[i,j] = \sum_{k_1,\cdots,k_{n-1}} \mathbf{A}_{i,k_1}\mathbf{A}_{k_1,k_2}\cdots\mathbf{A}_{k_{n-1},j}. \tag{16}$$

Note that $(i, k_1, \cdots, k_{n-2}, k_{n-1}, j)$ is a random walk with length $n$ if $\mathbf{A}[i,k_1] = \mathbf{A}[k_1,k_2] = \cdots = \mathbf{A}[k_{n-1},j] = 1$. Therefore,

$$\mathbf{A}^n[i,j] = \sum_{k_1,\cdots,k_{n-1}} \mathbf{1}_{(i,k_1,\cdots,k_{n-2},k_{n-1},j)\text{exists}}. \tag{17}$$

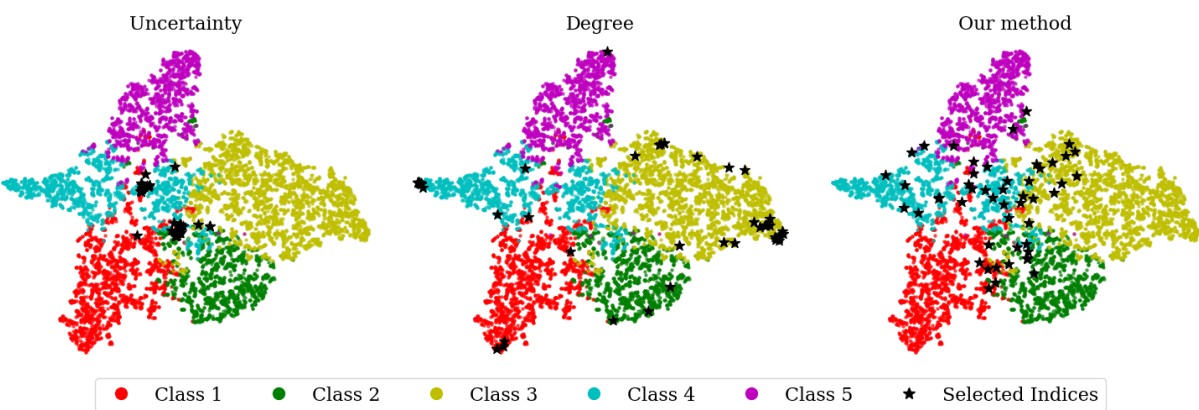

Figure A6: DBLPv7→ACMv9: visualization of logits scores output by the domain adaptation model on the target domain using t-SNE, with asterisks indicating the 50 nodes for active learning.

We set the message passing matrix in Eqn. 5 to be $\mathbf{S}^L = \sum_{n=0}^{L} e^{-\frac{E_n}{T}} \mathbf{A}^n$, which directly influences the path-oriented subnetwork. Then, we have

$$\mathbf{S}^L[i,j] = \sum_{n=0}^{L} e^{-\frac{E_n}{T}} \sum_{k_1,\cdots,k_{n-1}} \mathbb{1}_{(i,k_1,\cdots,k_{n-2},k_{n-1},j)\text{exists}}. \tag{18}$$

From this, we can tell that every path with the same length contributes equally to the message passing matrix $\mathbf{S}^L$, and the weight for paths of length $n$ is $e^{-\frac{E_n}{T}}$, which finishes our proof. $\qquad\square$

