# OpenReview forum: "DELTA: Dual Consistency Delving with Topological Uncertainty for Active Graph Domain Adaptation"
_TMLR — Accepted by TMLR_

### Review · Reviewer_czq2 · 2024-11-29

**Summary Of Contributions:**

This paper introduces a novel method for graph domain adaptation tasks. This approach leverages two complementary subnetworks, an edge-oriented graph subnetwork and a path-oriented graph subnetwork, and two loss functions, domain discrepancy measurement and final score computation.

**Audience:**

Yes

**Claims And Evidence:**

Yes

**Requested Changes:**

Please answer the questions above, especially Q1 and Q2.

**Strengths And Weaknesses:**

Strengths:

Achieves competitive scores compared to state-of-the-art methods.

Intuitive and well-designed experiments.

Questions and Weakness:

Q1: Chapter 4.2.1 describes the edge-oriented graph subnetwork, and Chapter 4.2.2 explains the path-oriented graph subnetwork. What is the difference between these two structures when n=1 for path-oriented graph subnetwork? Is the update formula combining Eq. (3) and Eq. (4) a special case of Eq. (5)? Is this a redundancy between these two structures?


Q2: The paper introduces a dual-structure approach without explicitly illustrating the advantages of this structure over using only one graph.  Table 2 includes an ablation study with two edge-oriented graphs or two path-oriented graphs. However, it is unclear whether using a single graph (edge-oriented or path-oriented) would produce similar results.
You can provide comparisons on one dataset to demonstrate the advantages of dual-structure graphs here, and you should discuss more in the camera-ready version.


Q3: In Table 1, "random" appears to refer to randomly selecting labels. Can you explain why "random" achieves surprisingly high F1 scores? I see GIFI[1] has lower scores in several datasets (e.g., A→C macro/micro, A→D micro)?

Q4: Missing GIFI[1] Scores in Table 1 since the paper mentions GIFI is currently the latest and most advanced graph domain adaptation model.

[1] Ziyue Qiao, Meng Xiao, Weiyu Guo, Xiao Luo, and Hui Xiong. Information filtering and interpolating for semi-supervised graph domain adaptation. Pattern Recognition, 153:110498, 2024.

---

### Review · Reviewer_jbPu · 2024-12-14

**Summary Of Contributions:**

This paper studies active graph domain adaptation problem where there is 1) a partially labelled source graph and 2) an unlabelled target graph. The aim is identifying a set of nodes on target graph to acquire the true labels for under a limited budget and thus enhancing the overall classification performance on target graph. The proposed model, DELTA, consists of two sub-networks, 1) an edge-based and 2) a path-based GNN, learned by existing supervision of  labelled source nodes and evaluated both on source and target domains. The estimated logits are utilized to identify a set of candidate nodes on target domain that have high inconsistency between two sub-networks. The evaluated inconsistencies are further adjusted by neighbourhood sizes of nodes at different hobs.  The domain discrepancy is evaluated by comparing the source and target domain on node feature space, but the structural properties are again incorporated by first-hob neighbourhood sizes of source nodes. DELTA is evaluated on a set of benchmark datasets and shown to perform better than active graph domain adaptation baselines.

**Audience:**

Yes

**Claims And Evidence:**

Yes

**Requested Changes:**

- I think for this type of work it is very important to share implementation, so would be great if authors can provide source code as supplementary material or share a public repository at camera-ready version if accepted.
- The baseline methods can to be broadened to include ideas from state-of-the-art graph domain adaptation methods. The active learning counterpart of following method could be important to compare to if possible:
    - Ziyue Qiao, Xiao Luo, Meng Xiao, Hao Dong, Yuanchun Zhou, and Hui Xiong. Semi-supervised domain adaptation in graph transfer learning. arXiv preprint arXiv:2309.10773, 2023.
- A theoretical explanation is needed to distinguish the contributions of edge-oriented and path-oriented subnetworks to topological modelling  would be insightful.
- Minor:
    - It would be useful to formalize the inverse square root of the normalization matrix in Equation (5) to improve the precision and reproducibility of the method.
    - I believe this paper is listed twice:

        Ziyue Qiao, Xiao Luo, Meng Xiao, Hao Dong, Yuanchun Zhou, and Hui Xiong. Semi-supervised domain adaptation in graph transfer learning. arXiv preprint arXiv:2309.10773, 2023a.

        so the list of references may need a double check.

**Strengths And Weaknesses:**

Strengths

- Studying graph domain adaptation in active learning settings is an important attempt as it could have high relevance to many applications at industrial level.
- The proposed model, DELTA, is intuitively strong and innovative.
- The ablation studies and sensitivity analysis are rich, and the interpretation of results is insightful.
- The paper is exceptionally well-written.

Weaknesses

- The importance and innovation in studying graph domain adaptation in active learning settings is not emphasized enough.
- The distinction between the theoretical roles of edge-oriented and path-oriented subnetworks is slightly unclear to me. Specifically, for any given node a two-layer MPNN as edge-oriented subnetwork involves information from neighbours at a distance of two at the final layer, like a path oriented subnetwork with $L=2$ in Equation (5). Existing GNNs can be viewed as a special case of path oriented convolution [1]. Although empirically shown to perform well, this raises questions about specific usage of edge and path oriented networks for discrepancy measurement at a theoretical level.
- Many state-of-the-art graph domain adaptation models do not operate on active learning setting, thus may not be directly comparable to the proposed model. Having said that integration of various backbones used for domain adaptation for comparison purposes could be relevant.
- Given the fact that the proposed method is only empirically proven to perform better than existing methods, it is very important to provide source code. Though I could not locate a public repository or supplementary material that could ensure reproducibility.

[1] Zheng Ma, Junyu Xuan, Yu Guang Wang, Ming Li, and Pietro Lio. Path Integral Based Convolution and Pooling for Graph Neural Networks. arXiv preprint arXiv:2006.16811, 2020.

---

### Review · Reviewer_6FHH · 2024-12-14

**Summary Of Contributions:**

The summary of the contributions of the paper is as follows:

- The paper addresses active graph domain adaptation, focusing on improving performance on target graphs with limited semantic information.

- The problem is complicated by topological complexity and distribution discrepancies across source and target graphs.

- The paper introduces *DELTA* (Dual Consistency Delving with Topological Uncertainty), a novel approach combining two encoding pathways to select informative target nodes by ensuring consistency across both subnetworks:
  - **Edge-Oriented Subnetwork**: Uses message-passing to capture neighborhood information.
  - **Path-Oriented Subnetwork**: Explores high-order relationships by analyzing substructures.

- It then aggregates local semantics from the K-hop subgraph of each node, weighted by node degrees, to estimate topological uncertainty. Additionally it compares target and corresponding source nodes to compute discrepancy scores for finer selection. The two scores are added for the final selection of target node selection for updating the model parameters and iterate.

- experiments show DELTA outperforms existing state-of-the-art methods on citation 3 networks.

**Audience:**

Yes

**Broader Impact Concerns:**

There are no "Broader Impact Concerns" for this paper.

**Claims And Evidence:**

Yes

**Requested Changes:**

Based on the aforementioned strengths and weaknesses, the requested changes are provided below:

- Overall, it is elusive from time to time to follow which of the two source or target graphs the authors refer to when they talk about nodes. Lack of proper notation and elaboration. For instance 4.2.3 it is unknown until the end of the section when it becomes clear the two sub-networks are applied to the target graph. Clarification and disentanglement is needed.

- in Algorithm 1, it is unclear why step 2 and 3 are applied to the source graph. It is not clear how s_each and s_path on the source graph is utilized in the algorithm down the road. Elaboration is needed.

- In section 5.5.1, the motivation behind the choice of the graph encoder backbones is not clear.

- In the experimental setup, the information on the number of iteration and the procedure to predict the labels on the target graph is missing. Please fill out the missing information.

- What does the following mean "active learning in a one-shot manner". Elaboration is helpful to help the reader follow the methodology.

- In section 5.1.3, what is GIFI model? Additionally, for reproducibility purposes details on the hyper-parameters of the GCN and PAN encoders are missing (e.g. number of layers, number channels etc). Additionally, open sourcing the code will help the community to expand the methodology.

- "we one-shot select the top" needs correction in 4.1.

- What is consistency fusion in Figure 2? where is it explained? Additionally, the figure may need to be revisited to clarify if the source graph is the input to the two pathways?

**Strengths And Weaknesses:**

## Strengths
- The paper is very well structured and procedurally developed to explain the methodology. The modularity of the methodology is very well maintained. The formulation of the idea is properly expressed in a simple manner that helps the reader to easily follow the paper.

- The literature review is instructive and insightful.

- The results on the ablation studies and parameter sensitivity sections are supportive.

## Weakness
- The major weakness is a lack of empirical evidence to support the hypothesis. It is approaching the lower bound to verify if the hypothesis is indeed supported according to the argued claims. Stronger and more convincing empirical studies and analysis would indeed provide more confidence to validate the hypothesis.

- There are some unclear aspects of the methodology that needs elaboration and justification. They are mentioned in the "Requested Changes" section.

- In section 4.2.3, the motivation behind why a Euclidean measure of distance between the two pathways is informative for the selection of the candidate nodes on the target graph is missing. It is not clear why the measure leads into "richer information and inconsistent prediction". It seems subjective and lacks proper qualitative definition and abstraction. Additionally, it is not clear if this process is applied to the source or target graphs or both and then how they are used later for the downstream modules in the framework.

---

### Decision · Action_Editor_EMCp · 2025-02-03

**Recommendation:** Accept with minor revision

**Comment:**

The paper works on active graph domain adaptation and proposes a method called DELTA that uses two encoding pathways to select informative target nodes by leveraging their inconsistencies, where the edge-oriented subnetwork/pathway uses message-passing to capture neighborhood information, and path-oriented subnetwork/pathway makes use of high-order relationships by analyzing substructures. It aggregates local semantics from the K-hop subgraph of each node, weighted by node degrees, to estimate topological uncertainty. Experiments show the efficacy of the proposed DELTA.

All three reviewers questioned the less convincing empirical evidence supporting claims made in the paper, in particular the (empirical and/or theoretical) evidence justifying the parallel design of the edge- and path-oriented subnetworks for complementary effects. The authors responded by 1) adding ablation studies, where results show that use of both the two subnetworks slightly improves the performance, and 2) a theoretical analysis on paths of different lengths contribute differently on learning (Theorem 4.1 in the revised paper). Reviewers are also concerned with other points including the less clear technical presentation of the method, the necessary motivation on the importance of the studied problem, and why the very baseline of the Random method performs quite well. The authors responded with clearer presentations. Two of the three reviewers lean to accept while reviewer czq2 favors negatively. AE agrees with the reviewers on the point that the benefits of parallel subnetworks are not justified enough; however, given completeness of the presented method and empirically verified method, the paper can be accepted according to TMLR criteria. Given acceptance, the authors need to make public their implementation and codes, since the current link has expired already.

The authors please prepare the final version including all the revisions made in the review process, and an official webpage making public the implementation details and codes.

**Audience:**

Yes

**Claims And Evidence:**

Yes